# CONTEXTGEN: CONTEXTUAL LAYOUT ANCHORING FOR IDENTITY-CONSISTENT MULTI-INSTANCE GENERATION

**Ruihang Xu, Dewei Zhou, Fan Ma,** * **Yi Yang**
ReLER Lab, CCAI, Zhejiang University
{ruihangxu,zdw1999,yangyics}@zju.edu.cn
flowerfan524@gmail.com

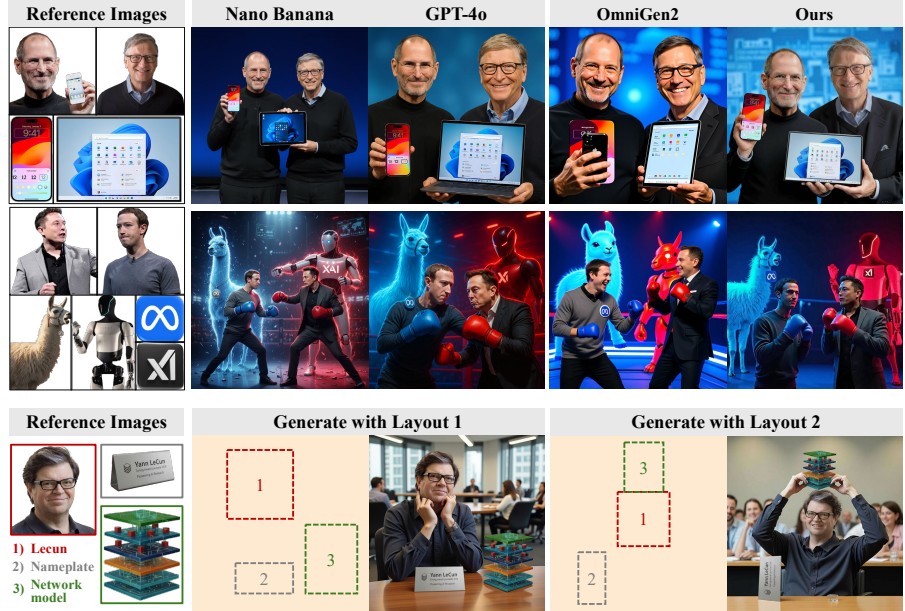

Figure 1: **Representative showcases of our work.** Upper panel: Our multi-subject-driven generation results versus existing open-source SOTA (OmniGen2) and proprietary models (Nano Banana, GPT-4o). Lower panel: Our layout-to-image generation examples using different layouts.

## ABSTRACT

Multi-instance image generation (MIG) remains a significant challenge for modern diffusion models due to key limitations in achieving precise control over object layout and preserving the identity of multiple distinct subjects. To address these limitations, we introduce **ContextGen**, a novel Diffusion Transformer framework for multi-instance generation that is guided by both layout and reference images. Our approach integrates two key technical contributions: a **Contextual Layout Anchoring (CLA)** mechanism that incorporates the composite layout image into the generation context to robustly anchor the objects in their desired positions, and **Identity Consistency Attention (ICA)**, an innovative attention mechanism that leverages contextual reference images to ensure the identity consistency of multiple instances. To address the absence of a large-scale, high-quality dataset for this task, we introduce **IMIG-100K**, the first dataset to provide detailed layout and identity annotations specifically designed for Multi-Instance Generation. Extensive experiments demonstrate that ContextGen sets a new state-of-the-art, outperforming existing methods especially in layout control and identity fidelity. Our code is available at https://github.com/nenhang/ContextGen.

---

* Corresponding Author.

# 1 INTRODUCTION

Diffusion-based models (Ho et al., 2020) have significantly expanded the horizons of image customization, with many recent systems (e.g., FLUX (Labs, 2024b)) adopting the Diffusion Transformer (DiT) (Peebles & Xie, 2022) framework for its enhanced generation quality. Recent developments in subject-driven image generation, such as OmniGen2 (Wu et al., 2025b), and layout-to-image synthesis, exemplified by MS-Diffusion (Wang et al., 2025), have further broadened the scope of customization, enabling control over both content and composition in generated images.

However, current methods face three fundamental limitations: **(1) Inadequate position control**, where existing layout guidance fails to achieve accurate spatial precision for user-specified arrangements; **(2) Weak identity preservation**, as subject-driven approaches struggle to maintain fine details across multiple instances, particularly with an increasing number of reference images. **(3) Lack of high-quality training data**, as existing datasets do not provide large-scale, precisely aligned pairs of reference images and layout annotations for multi-instance scenarios. These deficiencies collectively hinder the simultaneous achievement of compositional accuracy and identity fidelity.

To address these challenges, we propose **ContextGen**, a novel DiT-based framework that enables multi-instance generation by unifying two key modalities. **First**, we use a **composite layout image** for precise spatial control. As shown in the setup stage of Fig. 2, this layout image can be either user-provided or automatically synthesized. **Second**, we integrate **reference images** to overcome the limitations of layout-only generation, such as instance information loss due to overlaps. By incorporating these modalities into a unified **contextual** framework, ContextGen achieves both precise spatial control and high instance-level identity consistency.

Our work introduces three key innovations and contributions: **(1) Contextual Layout Anchoring (CLA)**, which leverages contextual learning to anchor each instance at its desired position by incorporating the layout image into the generation context, thereby achieving robust layout control; and **(2) Identity Consistency Attention (ICA)**, a novel attention mechanism which propagates fine-grained information from contextual reference images to their respective desired locations, thereby preserving the identity of multiple instances. Complementing these mechanisms is an enhanced position indexing strategy that systematically organizes and differentiates multi-image relationships. **(3) A large-scale, hierarchically-structured dataset, IMIG-100K**, which we curate with annotated bounding boxes and identity-matched references to directly address the current data scarcity in **I**mage-guided **M**ulti-instance **I**mage **G**eneration, with hierarchical samples shown in Fig. 3.

Our method achieves state-of-the-art performance across three benchmarks. On *(1) COCO-MIG* (Zhou et al., 2024b), it improves instance-level success rate by +3.3% and spatial accuracy (mIoU) by +5.9% over prior art. For *(2) LayoutSAM-Eval* (Zhang et al., 2024), it attains the highest scores in texture and color fidelity, demonstrating superior detail preservation. Most notably, on *(3) LAMICBench++* (Chen et al., 2025b), our approach outperforms all open-source models by +1.3% average score and even surpasses commercial systems like GPT-4o in identity retention (+13.3%). These gains validate CLA's layout robustness and ICA's effectiveness in multi-instance scenarios.

In summary, our key contributions are as follows:

- **ContextGen**: A novel DiT-based framework with **Contextual Layout Anchoring (CLA)** for robust layout control and **Identity Consistency Attention (ICA)** for precise identity preservation.
- **IMIG-100K**: The first large-scale, high quality, and hierarchically-structured dataset for image-guided multi-instance generation, which provides detailed layout and identity annotations.
- **SOTA Performance**: We achieve state-of-the-art results, outperforming existing methods (both open-source and proprietary) especially in layout control and identity preservation.

# 2 RELATED WORK

## 2.1 DIFFUSION MODELS

Diffusion models have evolved from UNet architectures (Ho et al., 2020; Rombach et al., 2022) to transformer-based approaches like DiT (Peebles & Xie, 2022), enabling scalable multimodal generation as seen in Stable Diffusion 3 (Esser et al., 2024). FLUX (Labs, 2024b) further advanced this by unifying visual and textual inputs through multi-modal attention mechanism.

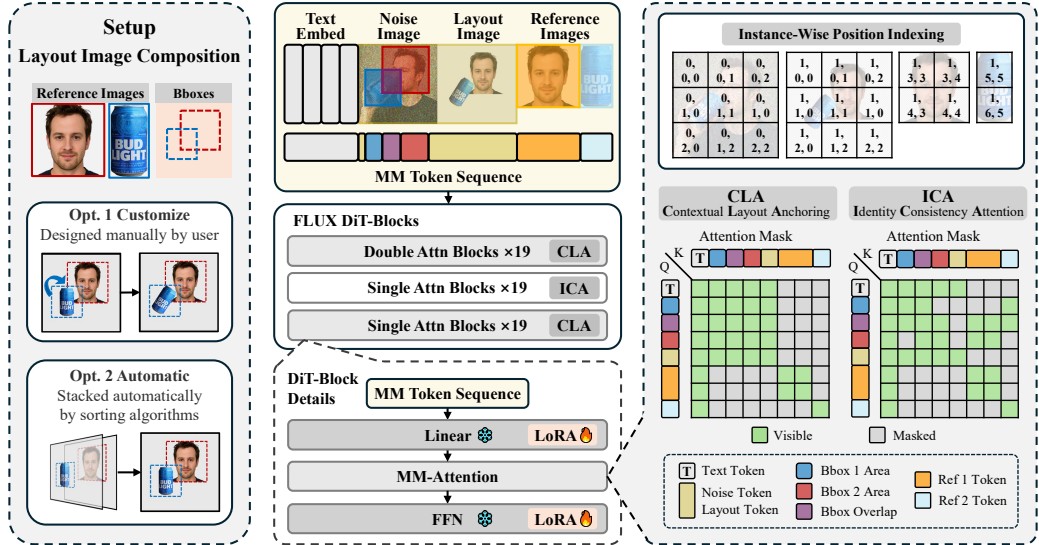

Figure 2: **Overview of ContextGen.** Left (Setup Stage): Options to composite the Layout Image. Middle (Model Core): Central generation architecture using FLUX DiT-Blocks. Right (Attention Mechanisms): Details of MM-Attention components (Position Indexing, CLA and ICA).

## 2.2 INSTANCE-LEVEL CONTROLLABLE IMAGE GENERATION

GLIGEN (Li et al., 2023) pioneered the layout-to-image generation paradigm. Follow-up studies utilizing UNet-based methods like InstanceDiffusion (Wang et al., 2024) and MIGC (Zhou et al., 2024a), or DiT-based approaches like EliGen (Zhang et al., 2025a) and 3DIS (Zhou et al., 2024c), have demonstrated enhanced capabilities in handling multiple instances. Current state-of-the-art frameworks like OmniGen2 (Wu et al., 2025b) and DreamO (Mou et al., 2025) process multi-subject conditions via integrated token sequences but face identity degradation with many subjects. While MS-Diffusion (Wang et al., 2025) and LAMIC (Chen et al., 2025b) combine reference-driven generation with layout control, challenges remain in layout precision and identity consistency.

## 3 METHOD

### 3.1 PRELIMINARIES

**Multimodal Diffusion Transformers (MM-DiT)** Recent architectures have replaced modality-specific cross-attention with unified multimodal processing. The MM-Attention operation concatenates image tokens $t_{\text{image}}$ and text embeddings $t_{\text{text}}$ into a single sequence $\mathbf{T} = [t_{\text{text}}, t_{\text{image}}]$, enabling joint self-attention across modalities. Stable Diffusion 3/3.5 (Esser et al., 2024; stability.ai, 2024) and FLUX (Labs, 2024b), treat all modalities within a shared latent space. The framework naturally supports in-context learning by allowing arbitrary interleaving of visual and textual tokens, while maintaining stable gradient flow across modalities during end-to-end training.

**Position Indexing and Attention Mask in MM-Attention** To address the permutation-invariance of the Transformer architecture, Rotatory Position Embedding (RoPE) (Su et al., 2023) was introduced to encode relative positional information. Adapting this for a unified multimodal space, the FLUX.1-Dev architecture proposes a novel extension of RoPE that employs a ternary position encoding scheme. This scheme assigns a position index $\mathbf{p}_i = (m, i, j)$ to each token in the sequence. The first component $m$ is set to 0 and is retained for further use. For text tokens, the spatial coordinates $(i, j)$ are fixed at $(0, 0)$, while for image tokens, they correspond to the spatial coordinates $(i, j)$ in the 2D noise latent space. This set of position indices $\{\mathbf{p}_i\}$ for the sequence forms a position index matrix $\mathbf{P}$.

The unified attention mechanism, which is controlled by the attention mask $\mathbf{M}$, integrates this positional information through RoPE. Specifically, the rotation matrix $\mathbf{R}$ is computed by applying the RoPE formulation to the position index matrix $\mathbf{P}$, a process we denote as $\mathbf{R} = \text{Rotate}(\mathbf{P})$. This

resulting matrix $\mathbf{R}$ is then utilized to apply a rotation to the query ($\mathbf{Q}$) and key ($\mathbf{K}$) embeddings before the dot-product calculation. The attention is then calculated as:

$$\text{MM-Attn}(\mathbf{Q}, \mathbf{K}, \mathbf{V}) = \text{softmax}\left(\frac{(\mathbf{RQ})(\mathbf{RK})^\top}{\sqrt{d}} \odot \mathbf{M}\right)\mathbf{V}, \tag{1}$$

where $\odot$ denotes element-wise multiplication. In the FLUX.1 series, a self-attention mechanism is employed where queries, keys, and values are all derived from the unified token sequence $\mathbf{T}$, and $\mathbf{M}$ is an all-True matrix, enabling full attention across all tokens.

## 3.2 Contextual Attention with Layout Anchoring and Identity Preservation

**Contextual Conditioning with Layout and Reference Images**    Recent studies in image-to-image (I2I) tasks have demonstrated the effectiveness of using a diptych, a side-by-side reference image pair, to guide diffusion models (Shin et al., 2024; Song et al., 2025; Zhang et al., 2025c). Building upon this, our framework introduces a novel layout control strategy by integrating a **composite layout image** into the generation context. This can be done either by a user-defined composition, which offers greater control and is often more aligned with specific user intent, or by our automated sorting algorithm (mentioned in Appx. A.2) based on the occlusion ratio of all instances. This composite diptych serves as the primary input for our **Contextual Layout Anchoring (CLA)** mechanism, which is designed to enforce a robust spatial structure by anchoring objects to their desired locations.

However, relying solely on this composite layout image presents challenges. As illustrated in Appx. B.5, in scenarios with instance overlap, the process of compositing may result in information loss or detail degradation. To address this, we integrate the original, high-fidelity reference images alongside the diptych. The unified token sequence $\mathbf{T}$ mentioned in Sec. 3.1 is constructed as:

$$\mathbf{T} = [\mathbf{t}_{\text{text}}, \mathbf{t}_{\text{image}}, \mathbf{t}_{\text{layout}}, \mathbf{t}_{\text{ref}_1}, \cdots, \mathbf{t}_{\text{ref}_N}]. \tag{2}$$

Our **Identity Consistency Attention (ICA)** mechanism incorporates these tokenized reference images $\{\mathbf{t}_{\text{ref}_i}\}$ into the context to preserve instance-specific attributes and details, effectively mitigating the issues of detail loss in overlapping regions, thereby ensuring a complementary relationship between the robust layout guidance from CLA and the precise detail preservation from ICA.

**Contextual Layout Anchoring (CLA)**    Inspired by the functional specialization observed in DiT layers (Zhou et al., 2025b; Zhang et al., 2024), we propose a hierarchical attention architecture to process the unified token sequence. As shown in the middle panel of Fig. 2, the **CLA** mechanism operates in the front and back layers, focusing primarily on global context and structural composition. The CLA mask, detailed in the right panel of Fig. 2, ensures broad communication across the text, image, and layout modalities. Using the token sets defined ($\mathcal{T} = \{\mathbf{t}_{\text{text}}\}$, $\mathcal{I} = \{\mathbf{t}_{\text{image}}\}$, $\mathcal{L} = \{\mathbf{t}_{\text{layout}}\}$, and $\mathcal{R}_n = \{\mathbf{t}_{\text{ref}_n}\}$) and reference bounding boxes $\{B_n\}_{n=1}^N$, the attention mask for CLA is defined as:

$$\mathbf{M}_{\text{CLA}}(q, k) = \mathbb{1}\left[(q, k) \in (\mathcal{T} \cup \mathcal{I} \cup \mathcal{L})^2 \cup \bigcup_{n=1}^N (\mathcal{R}_n \times (\mathcal{T} \cup \mathcal{R}_n))\right], \tag{3}$$

where $q$ and $k$ are arbitrary tokens from the query and key sequences respectively, and $\mathbb{1}[\cdot]$ denotes the indicator function.

**Identity Consistency Attention (ICA)**    While the front and back layers perform global spatial anchoring, we introduce the **ICA** mechanism in the middle layers to facilitate detailed, instance-level identity injection. As detailed in the right panel of Fig. 2, ICA operates by applying a specialized attention mask, $\mathbf{M}_{\text{ICA}}$, for tokens located within a specific bounding box. For a query token $q \in B_n$, the attention mask is defined as:

$$\mathbf{M}_{\text{ICA}}(q, k) = \mathbb{1}\left[(q, k) \in \bigcup_{n=1}^N (B_n \times (\mathcal{T} \cup B_n \cup \mathcal{R}_n)) \cup \{(q, k) \in \mathbf{M}_{\text{CLA}} \mid q \notin \bigcup_{n=1}^N B_n\}\right]. \tag{4}$$

The core function of $\mathbf{M}_{\text{ICA}}$ is the forced connection between $q$ and its corresponding reference tokens $\mathcal{R}_n$, ensuring reliable identity transfer. Tokens outside any bounding box (i.e., background) default to the mask used by CLA. This hierarchical strategy effectively transitions our framework from global layout control to refined instance-level identity preservation.

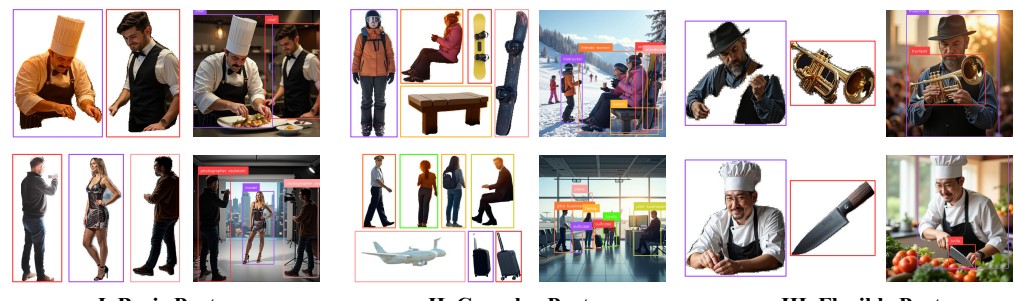

**I. Basic Part**  **II. Complex Part**  **III. Flexible Part**

Figure 3: **Image Samples of IMIG-100K Dataset.**

**Instance-Wise Position Indexing**  The ternary position encoding scheme described in Sec. 3.1 was extended in FLUX.1-Kontext (Labs et al., 2025) to handle image editing, where the first component of the position index, $m$, was set to 1 for edit tokens. Inspired by this work and other existing work (Wu et al., 2025c) that shows providing distinct and non-overlapping position indices for each image sequence significantly improves the model's ability to differentiate between various images, we propose a refined position encoding strategy to systematically structure the relationships within our unified token sequence $\mathbf{T}$ (Eq. (2)).

- **Basic Part:** The primary noise latent $\mathbf{t}_{\text{image}}$ retains the original $(0, i, j)$ indexing, ensuring spatial coherence within the target image.
- **Auxiliary Part:** Tokens from auxiliary inputs, including layout image and reference images, are assigned a unique index. They are indexed as $(1, W_n + i, H_n + j)$, where $W_n = \sum_{k=1}^{n-1} w_k$ and $H_n = \sum_{k=1}^{n-1} h_k$ are cumulative offsets aggregating the dimensions of all preceding conditioning images. This guarantees unique positional identifiers for each conditioning image, even when they are concatenated.

The effectiveness of this refined strategy is validated by our ablation study in Appx. B.2. This approach allows the attention mechanism to distinguish between tokens from the noise latent and auxiliary inputs, as well as to differentiate between tokens from various conditioning images.

### 3.3 IMIG-100K: AN IMAGE-GUIDED MULTI-INSTANCE-GENERATION DATASET

High-fidelity image-guided multi-instance generation is severely limited by the lack of suitable training data. While existing large-scale datasets (Lin et al., 2015; Deng et al., 2009) provide diverse instances, they often lack the aesthetic quality and annotation granularity required for modern diffusion models. Conversely, recent subject-driven datasets (Tan et al., 2025; Xiao et al., 2024a) exhibit high visual quality but are limited by their low instance multiplicity per image. A brief survey on related datasets is provided in Appx. C.1. To bridge this gap, we introduce **IMIG-100K**, a large-scale dataset created using the FLUX framework (Labs, 2024b). This dataset is specifically designed to support multi-instance generation by providing high-resolution, high-fidelity data with precise layout and reference images.

**Dataset Structure and Key Features**  To robustly train the diverse capabilities required for identity-consistent multi-instance generation, the IMIG-100K dataset is systematically structured into three specialized sub-datasets. These subsets collectively facilitate the comprehensive training of our framework, with examples shown in Fig. 3.

1. **Basic Instance Composition (50K samples):** This subset focuses on foundational compositional skills. The ground truth images are generated by the text-to-image model FLUX.1-Dev (Labs, 2024b), and we derive reference images using detection and segmentation models (Liu et al., 2023; Ravi et al., 2024; Dai et al., 2025). These reference images undergo minimal post-processing, including basic lighting adjustments.
2. **Complex Instance Interaction (50K samples):** Designed for more complex scenarios with up to 8 instances per image, this subset's data construction is similar to the basic part. However, the reference images are semantically edited to simulate real-world interactions, including occlusion, viewpoint rotation, and object pose changes.

3. **Flexible Composition with References (10K samples):** Unlike the previous two subsets, this unique subset is designed to train the model's robustness in handling low-consistency inputs. We first generate individual reference instances using the FLUX.1-Dev model. These are then composited into ground truth scenes by subject-driven models (Wu et al., 2025c; Mou et al., 2025), allowing for a much greater degree of flexibility and transformation in the composited instances relative to their original references. A key step involves rigorous filtering to ensure identity consistency from the references (Guo et al., 2021; Oquab et al., 2023).

All textual prompts are generated by advanced large language models (DeepSeek-AI, 2025; Comanici et al., 2025; OpenAI, 2024), ensuring diverse and high-quality descriptions. Further details on the dataset are provided in Appx. C.2.

## 4 EXPERIMENTS

### 4.1 EXPERIMENTAL SETTING

**Training Details**    We initialize the model with FLUX.1-Kontext (Labs et al., 2025) without introducing additional parameters and fine-tune it using LoRA (Low-Rank Adaptation) (Hu et al., 2021) with LoRA Rank 512. We perform training on 4× NVIDIA A100 GPUs with a total batch size of 16. The model is tuned on the three hierarchical sub-datasets described in Sec. 3.3 for 5K steps, employing the Prodigy optimizer (Mishchenko & Defazio, 2024) with its default learning rate. We also employ **Direct Preference Optimization (DPO)** (Rafailov et al., 2024) (detailed in Sec. 4.4) to refine text-visual alignment and user preference. These enable the model to evolve from mastering simple compositions to synthesizing complex multi-instance scenes.

**Benchmark Datasets**    We employ three distinct benchmark datasets for evaluation.

(1) **LAMICBench++**: A specialized benchmark for evaluating identity preservation and feature consistency in subject-driven generation. We extend the multi-image composition benchmark from LAMICBench (Chen et al., 2025b), aggregating multi-category reference images (humans, animals, objects, etc.) from established datasets including XVerseBench (Chen et al., 2025a), DreamBench++(Peng et al., 2025) and MS-Bench (Wang et al., 2025). In particular, we construct a dataset of 160 cases in total, including 50 cases with 2 reference images, 40 with 3, 30 with 4, 20 with 5, and 20 with over 5 reference images. These cases are divided into two categories: **Fewer Subjects** ($\leq 3$ reference images) and **More Subjects** ($\geq 4$ reference images). In this benchmark, we adapt and slightly modified the four evaluation metrics from the original work: (1) Global text-image consistency **(ITC)** evaluated through visual-question-answering (VQA) (Ye et al., 2024), with approximately 2K questions (4-12 per item); (2) Object preservation **(IPS)** (Liu et al., 2023; Oquab et al., 2023); (3) Facial identity retention **(IDS)** (Guo et al., 2021); (4) Aesthetic quality **(AES)** (Schuhmann, 2023).

(2) **COCO-MIG** (Zhou et al., 2024b): A benchmark designed to evaluate spatial and attribute accuracy in layout-to-image generation, comprising 800 images from COCO Dataset (Lin et al., 2015) with color-annotated instances. The evaluation metrics include: (1) Global and instance level success rate **(SR** and **I-SR)** determined by spatial accuracy **(mIoU)** and color correctness; (2) Multi-scale semantic consistency through global and local CLIP Scores **(G-C** and **L-C)**.

(3) **LayoutSAM-Eval** (Zhang et al., 2024): An open-set benchmark for layout-to-image evaluation, featuring 5K prompts with exhaustive entity-level annotations, from which we filter 1K samples with sufficiently large bounding boxes for reliable instance evaluation. We adapt the original work's metrics: (1) Fine-grained entity accuracy (**spatial**, **color**, **textural**, **shape**) evaluated using MLLM (Yao et al., 2024); (2) Holistic quality metrics: **CLIP** Score for semantic alignment and **Pick** Score (Kirstain et al., 2023) for human preference.

### 4.2 BASELINES

We compare our method against a comprehensive set of state-of-the-art baselines across relevant domains. For **layout-to-image generation**, we include pioneering works such as LAMIC (Chen et al., 2025b) and MS-Diffusion (Wang et al., 2025). To evaluate spatial control, we also benchmark against CreatiLayout (Zhang et al., 2024), EliGen (Zhang et al., 2025a), MIGC (Zhou et al., 2024b), 3DIS (Zhou et al., 2024c), InstanceDiffusion (Wang et al., 2024), and GLIGEN (Li et al., 2023). In the domain of **subject-driven generation**, we benchmark against OmniGen2 (Wu et al.,

Table 1: **Quantitative results on LAMICBench++**. Performance rankings: **bold** (highest), underline (second highest), wavy underline (third highest). The benchmark provides complete manual annotations for all method requirements: layout-aware methods (*) use our pre-annotated bounding boxes, while single-image-editing methods (†) use our manually composited layout images.

| Method | Fewer Subjects | | | | | More Subjects | | | | | AVG |
|---|---|---|---|---|---|---|---|---|---|---|---|
| | ITC | AES | IDS | IPS | AVG | ITC | AES | IDS | IPS | AVG | |
| LAMIC* | 42.27 | 50.26 | 37.02 | 74.17 | 50.93 | 28.29 | 50.84 | 24.63 | 60.87 | 41.16 | 45.61 |
| XVerse | 77.65 | 53.79 | 39.47 | 71.25 | 60.54 | 43.48 | 47.68 | 15.26 | 56.12 | 40.63 | 50.29 |
| MIP-Adapter | 87.22 | 56.50 | 6.63 | 68.40 | 54.69 | 71.88 | 58.38 | 1.12 | 61.10 | 48.12 | 51.28 |
| UNO | 89.86 | **58.04** | 17.53 | 75.34 | 60.19 | 77.25 | 58.90 | 7.83 | 62.94 | 51.73 | 55.58 |
| MS-Diffusion* | 89.13 | 57.67 | 12.45 | 75.49 | 58.69 | 78.46 | **59.65** | 9.06 | 69.75 | 54.23 | 56.35 |
| DreamO | 90.14 | 56.56 | 33.84 | 71.44 | 63.00 | 78.49 | 57.86 | 14.53 | 60.07 | 52.74 | 57.31 |
| Qwen-Image-Edit† | 93.63 | 57.97 | 17.71 | 73.30 | 60.65 | 86.35 | 59.57 | 9.32 | 65.26 | 55.13 | 57.57 |
| OmniGen2 | **95.40** | 57.58 | 32.17 | 73.14 | 64.57 | 89.69 | 58.49 | 15.15 | 69.31 | 58.16 | 61.08 |
| FLUX.1-Kontext† | 90.16 | 54.87 | **42.65** | 77.87 | 66.39 | **90.30** | 56.08 | 27.91 | 70.93 | 61.31 | 63.33 |
| Ours* | 92.54 | 57.50 | 35.86 | **81.23** | 66.78 | 89.89 | 59.18 | 30.42 | 73.35 | 63.21 | **64.66** |
| **Closed-Source Commercial Models** | | | | | | | | | | | |
| GPT-4o | **97.63** | **59.52** | 28.49 | 79.53 | 66.29 | 95.37 | **62.77** | 17.12 | 72.64 | 61.98 | 63.71 |
| Nano Banana | 96.58 | 58.48 | 34.36 | 80.87 | **67.57** | **95.48** | 60.81 | 16.67 | **74.11** | 61.77 | 64.11 |
| Ours* | 92.54 | 57.50 | **35.86** | **81.23** | 66.78 | 89.89 | 59.18 | **30.42** | 73.35 | **63.21** | **64.66** |

Figure 4: **Qualitative results on LAMICBench++.**

2025b), DreamO (Mou et al., 2025), UNO (Wu et al., 2025c), XVerse (Chen et al., 2025a), and MIP-Adapter (Huang et al., 2024) to specifically assess identity preservation. For a **cutting-edge benchmark**, we highlight the latest proprietary models, including Nano Banana (formally named Gemini 2.5 Flash Image, Google's latest multimodal model) (DeepMind, 2025) and GPT-4o-Image (OpenAI), as well as leading open-source models like Qwen-Image-Edit (Wu et al., 2025a) and FLUX.1-Kontext (Labs et al., 2025).

## 4.3 COMPARISON

**Identity Preservation and Overall Quality**   Quantitative results on LAMICBench++ in Tab. 1 show that our method excels in object preservation and facial identity retention. In Fewer Subjects, we achieve the highest IPS with competitive IDS. This advantage amplifies in More Subjects, while other open-source models experience significant drops in these metrics. Compared to closed-source models (GPT-4o and Nano Banana), we show a strategic trade-off: while slightly trailing in ITC and AES, we outperform them significantly in both IPS and IDS. This balanced performance yields our superior overall benchmark score (64.66 vs 63.71/64.11), demonstrating exceptional capability in preserving both objects and identities simultaneously.

Fig. 4 demonstrates our method's superior performance in preserving both content and style across diverse scenarios. Our approach consistently maintains accurate object relationships and fine details where other methods fail - evident in the precise rendering of facial identities (old man's wrinkles),

Table 2: **Quantitative results on COCO-MIG and LayoutSAM-Eval.** Image-guided methods (*) use our pre-generated images by FLUX.1-Dev (Labs, 2024b).

| Method | COCO-MIG Results | | | | | LayoutSAM-Eval Results | | | | | |
|---|---|---|---|---|---|---|---|---|---|---|---|
| | SR | I-SR | mIoU | G-C | L-C | Spatial | Color | Texture | Shape | CLIP | Pick |
| GLIGEN | 4.25 | 29.56 | 27.44 | 25.21 | 20.90 | 77.35 | 54.86 | 59.38 | 57.75 | 26.68 | 21.53 |
| LAMIC* | 1.25 | 13.56 | 21.17 | 21.82 | 18.71 | 77.27 | 69.04 | 69.96 | 68.74 | 23.49 | 21.91 |
| MS-Diffusion* | 4.50 | 28.22 | 34.69 | 25.50 | 20.77 | 85.41 | 73.94 | 76.08 | 75.21 | 26.92 | 22.22 |
| InstanceDiffusion | 23.00 | 60.28 | 54.79 | 25.77 | **21.91** | 86.39 | 71.39 | 76.73 | 75.37 | 26.36 | 20.96 |
| 3DIS | 18.88 | 55.44 | 49.35 | 23.72 | 20.40 | 88.34 | 80.97 | 82.52 | 81.30 | 26.75 | 21.89 |
| CreatiLayout | 19.12 | 54.69 | 48.96 | **26.22** | 20.70 | 93.59 | 77.43 | 79.62 | 78.89 | **27.99** | 22.44 |
| MIGC | 27.75 | 66.44 | 56.96 | 26.21 | 21.47 | 86.04 | 71.07 | 74.88 | 73.37 | 25.50 | 21.10 |
| EliGen | 26.00 | 64.12 | 59.23 | 24.92 | 20.58 | **94.05** | 83.84 | 87.31 | 87.01 | 26.89 | 22.27 |
| Ours* | **33.12** | **69.72** | **65.12** | 25.86 | 21.87 | 93.96 | **87.44** | **89.26** | **88.36** | 27.26 | **22.47** |

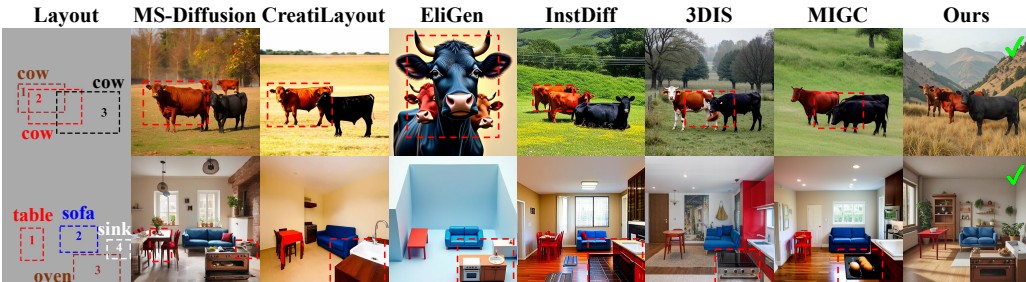

Figure 5: **Qualitative results on COCO-MIG.** We use red dashed box to indicate the missing, merged, dislocated and incorrectly attributed instances.

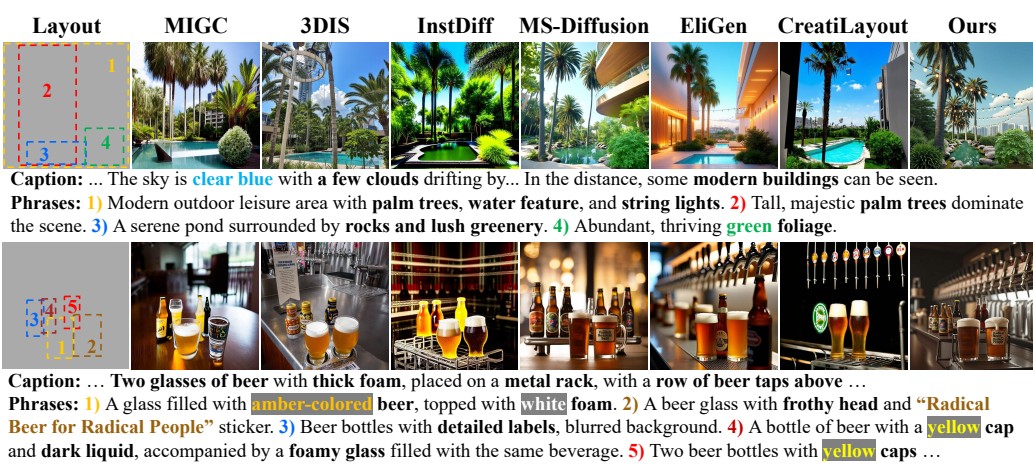

**Caption:** ... The sky is clear blue with **a few clouds** drifting by... In the distance, some **modern buildings** can be seen.
**Phrases: 1)** Modern outdoor leisure area with **palm trees**, **water feature**, and **string lights**. **2)** Tall, majestic **palm trees** dominate the scene. **3)** A serene pond surrounded by **rocks and lush greenery**. **4)** Abundant, thriving green **foliage**.

**Caption:** … **Two glasses of beer** with **thick foam**, placed on a **metal rack**, with a **row of beer taps above** …
**Phrases: 1)** A glass filled with amber-colored **beer**, topped with white **foam**. **2)** A beer glass with **frothy head** and **"Radical Beer for Radical People"** sticker. **3)** Beer bottles with **detailed labels**, blurred background. **4)** A bottle of beer with a yellow **cap** and **dark liquid**, accompanied by a **foamy glass** filled with the same beverage. **5)** Two beer bottles with yellow **caps** …

Figure 6: **Qualitative results on LayoutSAM-Eval.**

object features (shape of the vase, appearance of piggy bank, color and texture of Sphynx cat). More qualitative results are provided in Appx. D.1. Beyond fidelity, our model also exhibits high generative flexibility. A qualitative illustration of this adaptability is included in Appx. B.4, showcasing the model's ability to modify subjects' postures and attributes to comply with complex interactions given by text prompts (e.g., inter-subject interactions or dynamic scenes), thus proving it does not rigidly "transfer" the references.

**Layout Control and Attribute Binding**  Tab. 2 shows our method achieves superior layout control with the highest correctness on COCO-MIG. Direct comparison with text-guided L2I is infeasible due to differing input modalities, yet our image-guided approach provides more detailed and robust attribute binding. Crucially, compared to existing image-guided techniques, we lead in both layout fidelity and LayoutSAM-Eval color/texture accuracy.

Qualitative analysis, as presented in Fig. 5, highlights two key capabilities of our method. First, our approach effectively handles instance overlap, a common challenge for existing methods which often leads to attribute leakage or instance missing/merging. Second, our method exhibits superior spatial layout control, allowing it to synthesize a coherent and well-structured image from source images that may lack consistency. Additionally, as demonstrated in Fig. 6, our method performs robustly on complex text prompts, accurately reflecting fine-grained textual details in the generated image while preserving precise layout control. More qualitative results are provided in Appx. D.2 and D.3.

## 4.4 ABLATION STUDY

**Attention Mechanism Variations Across DiT-Blocks** We perform an ablation study to investigate the contribution of the ICA mechanism within our hierarchical attention architecture. We empirically divide the 57 DiT-blocks into three groups: FR-19 (first 19 blocks), MID-19 (middle 19 blocks), and BK-19 (last 19 blocks). The quantitative results on LAMICBench++ are summarized in Tab. 3.

Crucially, the baseline configuration (gray line in Tab. 3) that omits the CLA mechanism shows a substantial decline in performance across all metrics, reinforcing the indispensability of CLA for effective multi-instance generation.

Table 3: **Ablation study on applying ICA to different DiT-Blocks.** $F$, $M$, $B$ denote FR-19, MID-19, BK-19 blocks respectively. Gray line denotes the method w/o CLA.

| $F$ | $M$ | $B$ | ITC | AES | IDS | IPS | AVG |
|---|---|---|---|---|---|---|---|
| ✓ | ✓ | ✓ | 83.16 | 53.80 | 22.70 | 72.45 | 58.03 |
| ✓ | ✓ | ✓ | 91.54 | 58.41 | 24.19 | 74.46 | 62.15 |
| ✓ | ✓ |  | 91.14 | 57.36 | 26.08 | 74.17 | 62.19 |
|  |  | ✓ | 91.42 | 57.76 | 26.57 | 74.39 | 62.53 |
| ✓ |  |  | 91.20 | 57.00 | 30.80 | **77.63** | 64.16 |
| ✓ |  | ✓ | 90.26 | 58.35 | 31.27 | 76.99 | 64.22 |
|  | ✓ | ✓ | **91.55** | **58.85** | 31.10 | 75.64 | 64.28 |
|  | ✓ |  | 91.38 | 58.24 | **32.72** | 76.32 | **64.66** |

Prior work (Zhou et al., 2025b) has demonstrated that MID-19 blocks have the most significant influence on instance-specific attributes. In alignment with this finding, our experiments confirm that applying the ICA mechanism selectively to the MID-19 blocks yields the highest average score of 64.66 and the best IDS score of 32.72. Furthermore, the results in Tab. 3 suggest some potential functional focus across the FLUX-DiT layers, with a detailed analysis provided in Appx. B.6.

Importantly, while the value of the ICA may not be fully reflected by macro-level metrics alone, the ICA component is indispensable for preserving **fine-grained identity details** in scenes involving instance overlaps, as detailed in Appx. B.5.

**DPO Fine-tuning Analysis** To mitigate the model's tendency to rigidly copy layout images while neglecting instance adaptation (e.g., posture, lighting), we employ Direct Preference Optimization (DPO) (Rafailov et al., 2024), utilizing target images as preferred samples and layout images as less preferred. We first conduct an ablation study on LoRA Rank 256 to determine the optimal $\beta$ coefficient (results summarized in Tab. 4). The optimal $\beta$ (1000) was subsequently applied to our final DPO fine-tuning, which uses the higher capacity LoRA Rank 512. The qualitative validation for the final model is provided in Appx. B.3.3.

Table 4: **Ablation study on DPO $\beta$.**

| DPO $\beta$ | ITC | AES | IDS | IPS | AVG |
|---|---|---|---|---|---|
| 100 | 91.32 | **57.97** | 22.36 | 74.54 | 61.55 |
| 250 | **91.44** | 57.58 | 24.49 | 75.01 | 62.13 |
| 500 | 91.33 | 57.57 | 25.01 | 74.92 | 62.21 |
| 750 | 91.13 | 57.22 | 25.89 | 75.45 | 62.42 |
| 1000 | 91.03 | 57.10 | 26.83 | 75.71 | **62.67** |
| 1500 | 90.35 | 56.69 | 26.88 | 75.91 | 62.45 |
| w/o DPO | 86.84 | 54.19 | **32.37** | 76.78 | 62.55 |

Tab. 4 and further studies in Appx. B.3 reveals three key findings:

- **Improved Composition**: ITC and AES metrics consistently show improvement across tested $\beta$ values, demonstrating DPO's effectiveness in enhancing overall subject composition.
- **Controlled Trade-off**: Identity metrics (IDS and IPS) exhibit a slight and controlled decrease, with degradation scaling monotonically with the regularization strength $\beta$.
- **Optimal Configuration**: The setting of $\beta = 1000$ yields the highest overall average (AVG 62.67). This configuration successfully navigates the fidelity-flexibility trade-off, outperforming both the non-DPO baseline (62.55) and other tested $\beta$ settings.

This validates DPO's ability to navigate the fidelity-flexibility trade-off.

## 5 CONCLUSION

In this work, we presented **ContextGen**, a novel Diffusion Transformer-based framework for highly controllable multi-instance generation. The foundation of our approach is a unified contextual token sequence that integrates text, layout information, and multiple reference images, enabling a comprehensive understanding of the generation task. To tackle the twin challenges of precise spatial control and identity preservation, we introduced two dedicated mechanisms: the **Contextual Layout Anchoring (CLA)**, which effectively enforces robust spatial structure **using a composite layout image**, and the **Identity Consistency Attention (ICA)**, which ensures fine-grained instance-specific attribute preservation **via a constrained attention strategy**. Furthermore, our work involved an exploration of attention mechanism specialization across DiT layers, which informed the design of our hierarchical attention architecture. We also contributed the large-scale, hierarchically-structured dataset, **IMIG-100K**, complete with detailed layout and identity annotations, to accelerate future research in this domain. Extensive quantitative and qualitative evaluations confirm that ContextGen consistently outperforms state-of-the-art models, validating the efficacy and robustness of our proposed mechanisms. We believe this work provides a robust and scalable foundation for highly customizable image generation systems.

ACKNOWLEDGMENTS

This work was supported by National Natural Science Foundation of China (U2336212) and the Fundamental Research Funds for the Central Universities (226-202500080).

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

# A    MORE IMPLEMENTATION DETAILS

## A.1    BASE MODEL SELECTION

In our framework, different diffusion backbone models exhibit variations under fine-tuning-free settings for achieving high-fidelity multi-instance generation. We evaluated three variants of the FLUX family of models as potential backbones: FLUX.1-Dev (a general image generation model) (Labs, 2024b), FLUX.1-Fill (a local inpainting model) (Labs, 2024a), and FLUX.1-Kontext (an editing model) (Labs et al., 2025). While existing multi-subject-driven generation methods without layout control (Wu et al., 2025c; Hu et al., 2025) without attention masks have successfully utilized FLUX.1-Dev, our experiments showed a significant limitation: without additional fine-tuning, FLUX.1-Dev failed to produce coherent images when an attention mask was applied. In contrast, both FLUX.1-Fill and FLUX.1-Kontext demonstrated the ability to generate images correctly with the attention mask. Among these two, FLUX.1-Kontext exhibited a noticeably superior capacity for identity preservation. Therefore, we chose FLUX.1-Kontext as the backbone, as it fits better with our framework's attention-masking strategies and identity preservation goals. To isolate the impact of the backbone, we re-implemented a baseline method MS-Diffusion on FLUX.1-Kontext backbone for ablation studies in Appx. B.7. And we conducted qualitative comparisons with other state-of-the-art methods built upon equivalent FLUX-Family backbones in Appx. D.1.

## A.2    DETAILS OF COMPOSITING LAYOUT IMAGE

Our Contextual Layout Anchoring (CLA) mechanism relies on a meticulously constructed composite layout image to achieve robust spatial control. The design of this mechanism is conceptually inspired by the principles of multi-feature learning (Yang et al., 2013), which suggests that combining evidence from heterogeneous feature representations can lead to more robust semantic understanding. This process involves two key steps: determining the optimal composition order for all instances and then precisely placing each instance onto the canvas.

A correct composition order is crucial for multi-instance synthesis, especially when handling occlusions and complex overlaps. We propose a dynamic sorting algorithm, **Instance Layering Prioritization**, which first handles explicit containment relationships by prioritizing instances whose masks are completely contained within another's. For all other candidate instances, we use a **hybrid priority scoring system** to simulate the natural layering of objects. We utilize a pre-processing step to obtain each instance's precise effective area (Ravi et al., 2024). The priority score $P_i$ is calculated as:

$$P_i = \alpha \cdot \mathcal{A}(\text{instance}_i) + \beta \cdot \left( 1 - \sum_{j \neq i} \text{IoU}(\text{instance}_i, \text{instance}_j) \right) + \lambda \cdot \text{RandomFactor},$$

where $\mathcal{A}(\text{instance}_i)$ is the area of instance $i$, $\text{IoU}(\text{instance}_i, \text{instance}_j)$ is the Intersection over Union between instances, and $\alpha, \beta, \lambda$ are hyperparameters.

The proposed hybrid priority scoring system is designed to simulate general, high-probability composition orders for model training. The introduction of the random factor enhances data diversity and model robustness during training. Despite the Identity Consistency Anchoring (ICA) mechanism, the overlap relationships can still influence the final generated image. Thus, for inference, a user-provided layout offers a more direct and customized form of control.

# B    MORE DETAILS AND ANALYSIS ON EXPERIMENTS

## B.1    EXPERIMENTAL DETAILS ON LAMICBENCH++

As shown in Tab. 1, we benchmark our method against several strong baselines, including single-image-editing models (Wu et al., 2025a; Labs et al., 2025) and closed-source commercial models (OpenAI, 2024; DeepMind, 2025). Since these two categories of models require different input modalities and instruction methods, we prepared distinct inputs for a fair evaluation:

Table 5: **Ablation study on different position indexing strategies.**

| Position Indexing Strategies | ITC | AES | IDS | IPS | AVG |
|---|---|---|---|---|---|
| w/o x and y offsets | 90.53 | 57.92 | 21.54 | 74.57 | 61.14 |
| w/o y offsets | 90.22 | 57.60 | 21.01 | 74.02 | 60.71 |
| w/o x offsets | **91.53** | 57.87 | 26.71 | 74.59 | 62.67 |
| w/o editing token | 90.95 | 58.11 | 32.64 | 75.28 | 64.25 |
| w/ offsets and editing token | 91.38 | **58.24** | **32.72** | **76.32** | **64.66** |

**Single-Image-Editing Models**  These models are primarily designed for single-image editing, meaning they must process all instances combined within a single input image. We thus provided our manually composited layout images, ensuring minimal overlap across instances to avoid ambiguity. The following prompt template was used: "*Use the objects or humans in the image to create a new image that shows '*{PROMPT}*'. Preserve object features and human identities (if any, including facial details). You may fill in the background with appropriate details to achieve a natural and aesthetically harmonious result.*"

**Closed-Source Commercial Models**  These are general-purpose multi-modal models that accept multiple input images. To guide them to perform the specific multi-instance generation task while preserving identities, we relied on a dedicated prompt alongside the references. The prompt template was: "*Generate a high-quality image of '*{PROMPT}*'. Use the provided references, preserve object features and human identities (if any, including facial details).*"

## B.2 ABLATION ON POSITION INDEXING

As specified in Sec. 3.2, we extend the position indexing strategy from prior work (Wu et al., 2025c), which uses non-overlapping indices for explicit spatial delineation of objects, crucial for multi-instance distinction. The ablation results on LAMICBench++ are presented in Tab. 5. Our findings validate the importance of the positional encoding components:

- **The Necessity of Explicit Spatial Separation**: The results strongly confirm the importance of explicit spatial separation achieved via $x$ and $y$ offsets. Removing any offset component (e.g., *w/o x and y offsets*) significantly degrades all metrics, confirming the necessity of these offsets for the non-overlapping indexing mechanism. Its superiority is evidenced by the large margin observed in the IDS, suggesting better token distinction across different reference images in token sequence.

- **Role of Editing Token**: Our complete strategy includes an "editing token" (setting the first component of the position index triplet to 1). The ablation *w/o editing token* (setting it to 0) shows minimal performance difference in the final metrics. However, omitting this token actually resulted in a certain degree of gradient instability during the initial phase of training, which justifies its inclusion for a more stable optimization process.

## B.3 DETAILS OF DPO FINE-TUNING

### B.3.1 QUALITATIVE RESULTS OF DPO FINE-TUNING PROCESS

The visualization in Fig. 7 illustrates the efficacy of Direct Preference Optimization (DPO) in enhancing image generation, particularly by mitigating the issue of rigidly copying the layout image with blank backgrounds. To demonstrate this, we intentionally select an input and seed configuration that typically results in a minimal background scene.

The initial result on the left of Fig. 7a correctly anchors the main subject but suffers from the blank background issue, failing to render any environmental context. As the fine-tuning process advances, the model begins to introduce more naturalistic details, first by generating a realistic shadow and subsequently by adding a simple yet coherent background. Upon convergence, the final image on the right features a rich, detailed background, effectively demonstrating DPO's ability to enrich the overall scene while strictly preserving the subject's layout.

Fig. 7b shows how the DPO $\beta$ parameter affects the generation quality. A high $\beta$ value may limit the model's capacity for meaningful tuning, whereas an overly low $\beta$ value can cause the model to

follow the preference data too aggressively, risking a loss of the subject's identity during convergence (as seen with the leather bag when $\beta = 50$). Our results validate that when $\beta$ is correctly calibrated, DPO substantially improves image quality with minimal compromise to the subject's identity.

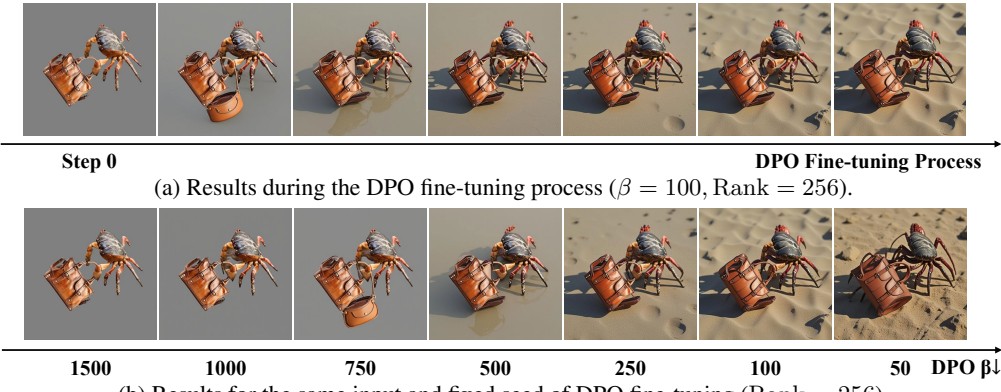

**Step 0**                                           **DPO Fine-tuning Process**

(a) Results during the DPO fine-tuning process ($\beta = 100$, $\mathrm{Rank} = 256$).

  **1500**        **1000**       **750**       **500**       **250**       **100**       **50**   **DPO β↓**

(b) Results for the same input and fixed seed of DPO fine-tuning ($\mathrm{Rank} = 256$).

Figure 7: **Results for the same input and fixed seed of DPO fine-tuning.**

### B.3.2 IMAGE QUALITY ASSESSMENT ON DPO FINE-TUNING

In order to fully assess the impact by DPO fine-tuning on image quality, we report the quality assessment results in Tab. 6. However, the FID score on LayoutSAM-Eval may not be highly referential, not only because FID is less sensitive to fine-grained image details, but also because the calculation process necessitates severe distortion by resizing the structurally diverse ground-truth images of this benchmark, thus failing to reflect the objective quality of the generated outputs. As shown in Fig. 8, the model with the best FID score on LayoutSAM-Eval exhibits a less satisfactory overall visual effect, specifically in background richness, light and shadow rendering, and fine detail processing, compared to the results obtained using a smaller DPO $\beta$ value.

To better assess the overall visual quality and fidelity of the generated images, despite the metrics AES and FID score from the benchmarks, we conducted a **user study**. In this user study, we sampled 20 sets of inputs and seeds from each of the two benchmarks (LAMICBench++ and LayoutSAM-Eval). For each sampled set, we generated results using 6 different DPO $\beta$'s, alongside a result from the model without DPO fine-tuning. This yielded a total of 7 images for comparison per set. 10 reviewers were asked to select **two** images with the best quality and **two** images with the worst quality in each group of images. Each "best quality" selection added one point to an image's score, while each "worst quality" selection deducted one point. The results are normalized and recorded at *User Preference* metric in Tab. 6. The comparison results further demonstrate that DPO fine-tuning will enhance image quality, and a smaller $\beta$ (a more aggressive DPO) generally leads to better quality.

Table 6: **Image quality assessment results on benchmarks and user study.**

| DPO $\beta$ | AES | FID | User Preference |
|---|---|---|---|
| 100 | **57.97** | 55.97 | **0.54** |
| 250 | 57.58 | 55.60 | 0.37 |
| 500 | 57.57 | **55.32** | 0.16 |
| 750 | 57.22 | 55.53 | 0.03 |
| 1000 | 57.10 | 55.65 | -0.11 |
| 1500 | 56.69 | 55.70 | -0.43 |
| w/o DPO | 54.19 | 55.93 | -0.56 |

### B.3.3 QUALITATIVE VALIDATION ON FINAL MODEL

Following the ablation study in Tab. 4, we applied the optimal DPO setting ($\beta = 1000$) to our final ContextGen model, which utilizes LoRA Rank 512. The qualitative results presented in Fig. 9

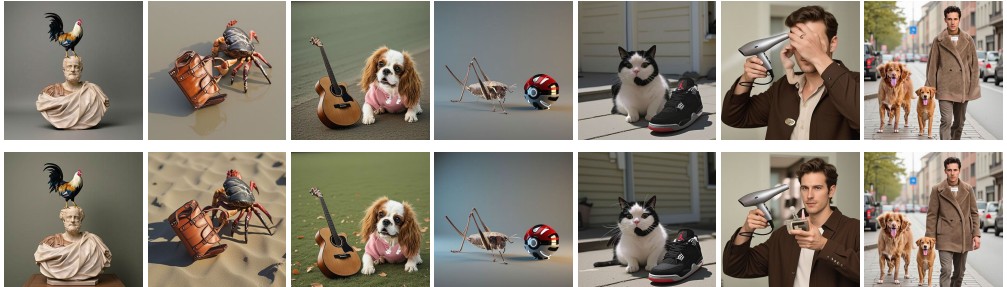

Figure 8: **Qualitative Comparison: DPO $\beta = 500$ (FID-Optimal) vs. DPO $\beta = 100$.** The comparison showcases the model fine-tuned with the FID-optimal setting (DPO $\beta = 500$, upper panel) versus a more aggressive setting ($\beta = 100$, lower panel). All corresponding images were generated with identical inputs and random seeds. The columns are specifically grouped to demonstrate the quality differences: Columns 1–3 highlight background richness; Columns 4–5 illustrate light-and-shadow rendering; and the final two samples reveal differences in fine identity details. Zoom for more details.

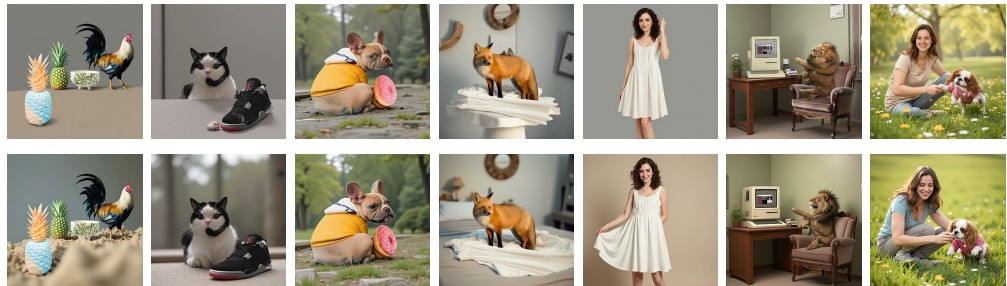

Figure 9: **Qualitative comparison of final Model ($\mathrm{Rank} = 512$): before vs. after DPO Fine-tuning.** The upper panel shows images generated before DPO fine-tuning, and the lower panel shows the results after. All corresponding images were produced using the same random seed.

strongly validate that DPO fine-tuning significantly enhances the visual quality and overall fidelity of the outputs from this high-rank configuration. This enhancement is particularly evident in several key aspects: increased richness in background details, improved flexibility in instance composition and more natural rendering of light and shadow.

### B.4 RESULTS AND ANALYSIS ON MODEL'S GENERATION FLEXIBILITY

The model's final output represents a crucial trade-off between Fidelity (adherence to reference image identities and given layout) and Flexibility (the ability to incorporate text descriptions, especially for inter-subject interactions). The richness of input text prompt significantly modulates this balance. When a simple or minimal prompt is provided, the model inherently prioritizes high fidelity, leaning towards preserving the exact details and posture defined by the reference images. Conversely, by enriching the prompt with complex descriptions and detailed interactions between subjects, the model will exhibit greater flexibility in the generation results.

Fig. 10 demonstrates how prompt complexity influences the generation outcome while the layout image remains constant. For instance, in the first panel of Fig. 10, when the action is described as "having a conversation", the model exhibits flexibility by removing the sword from the pixelated figure, a detail conflicting with the context. Furthermore, when the prompt is "in a fierce dual", the figure's posture is greatly modified to align with the dynamic description.

### B.5 ICA: AN INDISPENSABLE COMPONENT HANDLING IDENTITY IN OVERLAPPED REGIONS

Although a pure CLA method (using only a layout image as contextual input) exhibits scores similar to strategies with ICA in the aggregated results of Tab. 3, the critical necessity of the ICA component

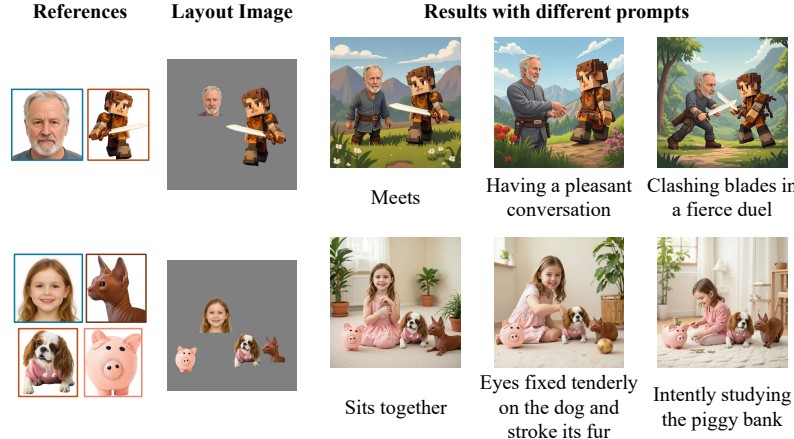

Figure 10: **Flexible compositions with different prompts.**

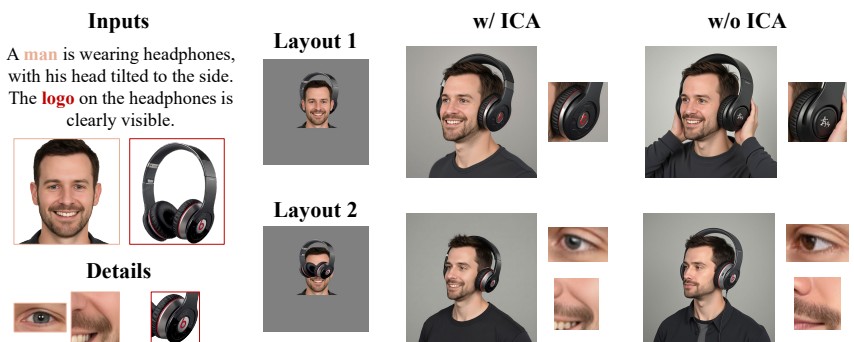

Figure 11: **Results of overlapping layout on methods with and without ICA.**

is explicitly demonstrated when handling layouts with overlaps. For generating layout involving overlapping instances, ICA is critical because the reference images in the token sequence provide extra identity information for the regions being occluded by other instances, mitigating identity loss in these overlapping areas.

Fig. 11 illustrates this effect across two scenarios: For Layout 1, subtle details like the red logo of the headphones are successfully preserved by the method with ICA but completely abandoned by the method without ICA. Conversely, in the occluded scene of Layout 2, the ICA method better retains fine-grained facial identity features—such as the color of the eyes and the presence of the nasolabial fold—compared to the non-ICA method, which loses these crucial details.

However, these overlapping cases are relatively rare in the current benchmark, and the differences on the image lead to a limited overall impact on the macro-level evaluation indicators. But these qualitative results demonstrate that ICA is an indispensable component for ensuring identity preservation in such cases. And its absence significantly impacts the final image's visual quality and detailed fidelity.

### B.6 ANALYSIS ON LAYER-WISE ICA ABLATION

As shown in Tab. 3, applying ICA to different layers (grouped by First-19, Mid-19 and Back-19) reveals an interesting result. Our ablation study revealed a gap between two distinct sets of strategies: the first set, comprising $\{F + M + B, F + M, B\}$, consistently yielded lower scores, while the second set, $\{F, \text{w/o CLA}, M + B, M\}$, achieved closely clustered higher scores. This result suggests the existence of intrinsic properties within the FLUX-DiT layers, indicating potential functional difference where performance is sensitive to the placement of attention mechanism.

- **F-19: Strategy to Minimize Modality Interference.** Prior work CreatiLayout (Zhang et al., 2024) extending the F-19 blocks has shown a powerful control in both spatial and attribute. And as

stated this work, no further modification is made to the other blocks (Single Stream Blocks, i.e., Mid-19 and Back-19 blocks) in order to reduce the potential interference among modalities (i.e., text, layout, and image). It does make sense in considering the Front-19 blocks contains a huge amount of parameters compared to the others. Adding ICA to these blocks is sufficient, while applying ICA to more blocks lead to some side effect like identity distortions. This aligns with our observation that $F + M + B$ and $F + B$ strategies underperform others.

- **M-19: Critical Role in Attribute Binding.** Prior work DreamRenderer (Zhou et al., 2025b) demonstrates through ablation study that Mid-19 blocks play a relatively more critical role in attribute binding. Our ablation study validates the finding, for applying ICA to Mid-19 blocks yields the best performance in our task on benchmark score.

- **F-19 Protection and Similar Performance.** The four strategies, $F$, $M$, $M + B$, and no ICA, yield relatively close performance scores because they all avoid posing further disruptions to the Front-19 blocks. And we can derive from this result that these four strategies do not have a very significant difference in overall performance.

- **B-19: Detrimental Impact on Final Refinement.** Conversely, applying ICA solely to the Back-19 blocks proves detrimental to performance, which is also aligned with the ablation study in DreamRenderer. For the Back-19 blocks serve as final and sensitive post-processing layers in each noise prediction step. Introducing attention mask exclusively at this late stage without any modifications to the earlier blocks will affect the refinement process, leading to problems like artifacts and blurring identity.

We choose Mid-19 from the last four strategies with very close score for the following reasons:

- **Performance**: Applying ICA to the Mid-19 blocks demonstrates an advantage on LAMICBench++ average score over other strategies.

- **Indispensability** of ICA: The analysis in Appx. B.5 has revealed the indispensability of ICA.

- **Efficiency**: Considering that the computational cost of applying the attention mechanism scales with both the number of blocks and the block capacity (such as F-19 blocks with much denser parameters), we selected the Mid-19 configuration for a relatively efficient approach to achieve better performance.

### B.7 MS-FLUX: A CONTROLLED CASE STUDY OF STRATEGY FAILURE ON STRONG BACKBONE

Previous image generation methods based on older diffusion models (e.g., Stable Diffusion) have shown considerable improvements over their base architectures. For instance, MS-Diffusion (Wang et al., 2025), built on SDXL, achieved image-guided layout control via its ***Grounding Resampler*** mechanism based on SD-Unet cross attention architecture. This Resampler fuses reference image content with explicit spatial and phrase information into multi-modal prompt tokens to guide generation.

To quantify the contribution of our ContextGen mechanisms and assess their synergistic fit with the FLUX backbone, we implement a competitor **MS-FLUX**. This implementation adapts a similar *Grounding Resampler* architecture onto the FLUX-DiT. We fine-tuned MS-FLUX with all the *Grounding Resampler*'s parameters trained and a LoRA Adapter (R=512) applied to the DiT blocks. After 50K training steps on IMIG-100K, we evaluate this model on COCO-MIG and LAMICBench++ for a comparison, the results of which are presented in Tab. 7. This result demonstrates **severely poor identity preservation** coupled with a **complete degradation of layout control** of MS-FLUX. The method of integrating reference images, phrases and layout information together as a text-prompt-like token sequence, which works on SD-Unet, works poorly on FLUX-DiT. In summary, this confirms that despite the strong contextual capability of FLUX.1-Kontext (FLUX-DiT) over old backbones, not all methods (even those that seem reasonable in principle or already worked on previous backbones, just like this *Grounding Resampler* in MS-Diffusion) will work on FLUX backbone. This further suggests that our specialized mechanisms provide a more correct design for achieving better layout control and identity-consistency on the FLUX backbone.

To ensure complete transparency regarding our comparative experiments and to fully document the controlled competitor, we include the implementation code details of MS-FLUX below. The code

Table 7: **Quantitative result of MS-FLUX on key metircs of LAMICBench++ and COCO-MIG.**

| Method | SR | I-SR | mIoU | IDS | IPS |
|---|---|---|---|---|---|
| MS-FLUX | 0.38 | 7.3 | 18.71 | 3.20 | 60.22 |
| MS-Diffusion | 4.50 | 28.22 | 34.69 | 9.06 | 69.75 |
| Ours | 33.12 | 69.72 | 65.12 | 32.72 | 76.32 |

involves three core stages: Resampler initialization, multimodal input encoding, and feature injection into the Transformer.

- **Resampler Initialization and Input Preparation.** The `Grounding Resampler` is initialized within custom `LightningModule` to match the dimensions of the FLUX Transformer. During the training step, image, layout, and phrase information are prepared.

```
# Initialization (in the LightningModule's setup method)
from .projection import Resampler
# ...
self.grounding_resampler = Resampler(
    dim=1280,
    depth=4,
    dim_head=64,
    heads=20,
    num_queries=512,
    embedding_dim=self.flux_pipe.transformer.config.in_channels,
    output_dim=self.flux_pipe.transformer.context_embedder.out_features,
    ff_mult=4,
    latent_init_mode="grounding",
    phrase_embeddings_dim=self.flux_pipe.text_encoder.config.projection_dim,
).to(self.target_dtype)
self.grounding_resampler.requires_grad_(True).train()
```

- **Encoding Reference and Spatial Information.** The `prepare_reference_latents` function encodes the reference images into latents. Concurrently, instance phrases and bounding boxes are encoded and assembled as "Grounding Keywords".

```
# Preparing inputs within the Custom Lightning Model's step() method
# ...
# 1. Extract and normalize boxes
phrases = [inst["phrase"] for inst in instance_info[0]]
boxes = [inst["bbox"] for inst in instance_info[0]]
boxes = torch.stack(boxes).to(self.device, self.target_dtype)
boxes = boxes / torch.tensor([width, height, width, height],
    device=self.device, dtype=self.target_dtype)

# 2. Encode Instance Phrases using FLUX's text encoder
phrase_input_ids = []
for phrase in phrases:
    # ... (Tokenizer call)
    phrase_input_ids.append(phrase_input_id)
phrase_input_ids = torch.stack(phrase_input_ids)
phrase_input_ids = phrase_input_ids.view(-1, phrase_input_ids.shape[-1])
phrase_embeds =
    self.flux_pipe.text_encoder(phrase_input_ids.to(self.device)).pooler_output

# 3. Assemble Grounding Keywords
grounding_kwargs = {
    "boxes": boxes,
    "phrase_embeds": phrase_embeds,
    "drop_grounding_tokens": [0],
}

# 4. Prepare reference image latents
```

```
reference_info_dict = prepare_reference_latents(
    # ... (calling prepare_reference_latents encodes the reference
        images into latents)
)
instance_latents = reference_info_dict["instance_latents"][0]
instance_latents = torch.cat(instance_latents, dim=0)
# ...
```

- **Resampler Processing and Feature Injection.** The `instance_latents` (visual features) and `grounding_kwargs` (spatial and phrase features) are passed to the Resampler. The Resampler's output is then concatenated into the token sequence input of the FLUX Transformer layer.

```
# Calling the Grounding Resampler
img_prompt_hidden_states = self.grounding_resampler(
    x=instance_latents,
    grounding_kwargs=grounding_kwargs,
)
img_prompt_hidden_states = img_prompt_hidden_states.reshape(bs, -1,
    img_prompt_hidden_states.shape[-1])

# Injecting the Resampler output into the custom Transformer forward
    pass
transformer_out = FluxTransformer2DModel_forward(
    self=self.flux_pipe.transformer,
    hidden_states=x_t,
    # ... (standard FLUX arguments)
    img_prompt_hidden_states=img_prompt_hidden_states, # <-- Resampler
        Output Injection
    txt_ids=text_ids,
    img_ids=img_ids,
    img_prompt_ids=img_prompt_ids,
    joint_attention_kwargs=None,
    return_dict=False,
)
```

- **Context Aggregation.** In `FluxTransformer2DModel_forward`, the Resampler's output (`img_prompt_hidden_states`) represents the fused image, bounding box, and phrase information. This feature vector is concatenated with the standard text (`encoder_hidden_states`) to form a unified, comprehensive context for the Transformer.

```
# Context Aggregation Logic (inside the custom
    FluxTransformer2DModel_forward)

# 1. Process standard text tokens
encoder_hidden_states = self.context_embedder(encoder_hidden_states)

# 2. Integrate the MS-FLUX Resampler Output
encoder_hidden_states = torch.cat((encoder_hidden_states,
    img_prompt_hidden_states), dim=1)

# 3. Update corresponding position indices
ids = torch.cat((txt_ids, img_prompt_ids, img_ids), dim=0)
image_rotary_emb = self.pos_embed(ids)

# ... (Encoder hidden states are then passed to the double transformer
    blocks)
```

- **Inference Time Injection.** During inference, the pre-calculated Resampler output is consistently passed into the Transformer at every denoising step.

```
# Inference logic in the custom pipeline forward function
# ...
noise_pred = (
```

```
    # ... (Standard Transformer call if reference_dict is None)
    if reference_dict is None
    else FluxTransformer2DModel_forward(
        self=self.transformer,
        hidden_states=latent_model_input,
        img_prompt_hidden_states=reference_dict["img_prompt_hidden_states"],
            # <-- Inference Injection
        timestep=timestep / 1000,
        # ... (other arguments)
        return_dict=False,
    )[0]
)
# ...
```

## C  MORE DETAILS ON MULTI-INSTANCE DATASET

### C.1  BRIEF SURVEY AND DISCUSSION ON EXISTING MULTI-INSTANCE DATASETS

Various approaches have been adopted to construct multi-instance datasets. Prior works like OmniGen series (Wu et al., 2025b; Xiao et al., 2024b) and MS-Diffusion (Wang et al., 2025) use a cross-verification method to match and pair group images and corresponding individual images from web images or video frames. And some methods begin to leverage synthetic data for training. For example, UNO (Wu et al., 2025c) employs a co-evolutionary data generation and training process using the model itself, and XVerse (Chen et al., 2025a) add data generated by FLUX.1-Dev as supplementary to high-aesthetic-quality images.

However, the existing datasets contain some limitations as robust training resources for spatial-aware multi-instance generation.

- **Insufficient Instance Count and Lack of Layout Annotations.** Though datasets from general-purposed generation methods or these above mentioned subject-driven works often exhibit a relatively high overall quality, identity consistency and a large overall scale, the data with more than 3 subjects is rather limited. Critically, none of these datasets provide detailed spatial or layout annotations.

- **Restricted Subject Variation between Reference and Target.** MS-Diffusion's dataset, while featuring more subjects per image and layout annotations, has a crucial limitation mentioned in the original paper: the low identity matching rate during cross-verification process across the video frames often necessitates using parts of the ground-truth image directly as reference image, which will limit the necessary variations and meaningful transform between the subjects in reference and target.

To address these limitations, we propose the pipeline in Sec. 3.3 to construct our own IMIG-100K dataset. We choose to use the synthetic data for the following reasons:

- **Real-World Data Scaling Challenges.** The collection and matching process in real-world data is relatively less flexible and more difficult to scale in both diversity and quantity.

- **Successful Precedent of Synthetic Data Usage.** The successful practice in previous work, such as UNO, has demonstrated the effectiveness of evolving synthetic data in training robust subject-driven generation models.

### C.2  MORE DETAILS OF IMIG-100K DATASET

Sec. 3.3 provides a general introduction to the synthesis and architectures of our dataset. More details and samples of our IMIG-100K Dataset are presented as follows.

**Data Diversity**  We have designed multiple prompt generation templates to include more scenes and styles. These templates are randomly combined and fed to LLMs maximize the diversity of the resulting image generation prompts. We also use different LLMs (i.e. DeepSeek v3 (DeepSeek-AI, 2025), Gemini 2.5 Flash (DeepMind, 2025) and GPT-4o (OpenAI, 2024)) for lexical and stylistic

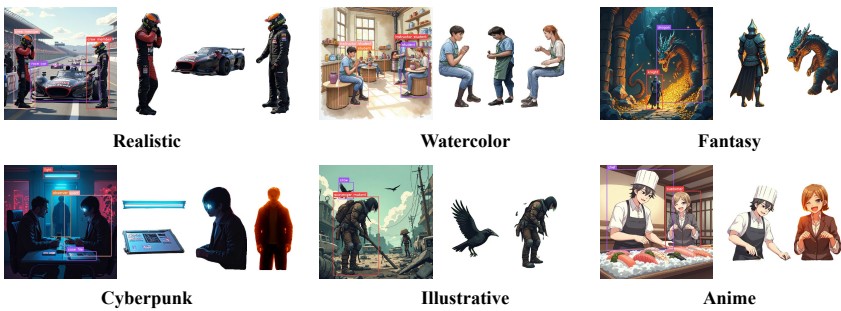

Figure 12: **Samples with multiple styles in IMIG Dataset.**

variations. Besides the realistic style (natural style), as illustrated in Fig. 12, we also include extra styles like fantasy and anime, which account for more user scenarios. The proportion of natural scenes remains high, accounting for approximately 85% to 90% of the total data.

**Detailed Spatial Annotations**  Every instance's is meticulously annotated with detailed spatial information, including the instance phrase, its valid area's bounding box and mask, and its corresponding position within the ground-truth image. The third subset of IMIG-100K—the flexible composite part—features more detailed annotations. Recognizing that many user scenarios only provide a facial avatar instead of a half or full-body image, we use face-related tools (Xin, 2022; Guo et al., 2021) to restore and standardize the faces in reference images into aligned facial images, as shown in Fig. 14. Face pairs in the ground-truth images (composited images) and corresponding reference images are strictly filtered by face quality and similarity. Crucially, the positions of valid faces are annotated (indicated by the green round bounding boxes in Fig. 14) across composited images, corresponding reference images and aligned face images. This enables a precise position and scale alignment when positioning and sizing reference instances in canvas in training process, instead of relying on a very rough bounding box of the entire instance.

**Overall Quality**  Our dataset has undergone strict quality assessment (Kirstain et al., 2023; Boutros et al., 2023) for both reference images and ground-truth images. And we leverage MLLMs to ensure the semantic consistency between reference images and generated captions. This aligns with recent efforts in multi-context understanding (Xu et al., 2025) and multimodal condition alignment (Zhou et al., 2025a), which both emphasize maintaining logical and semantic integrity across complex visual inputs. Additional samples illustrating the quality and annotation detail are provided in Figs. 13 and 14.

# D  MORE RESULTS AND DEMOS

## D.1  QUALITATIVE RESULTS COMPARING ACROSS FLUX-BASED METHODS

Several FLUX-based methods are included in our comparison on LAMICBench++: vanilla FLUX.1-Kontext, LAMIC (based on FLUX.1-Kontext), UNO and DreamO (both based on FLUX.1-Dev). Given that these competitors and our method base on highly similar (or identical) foundational backbones, we provide additional qualitative results in Figs. 15 and 16 for a direct and fair visual comparison.

## D.2  MORE RESULTS ON COCO-MIG

Full quantitative result on COCO-MIG benchmark is shown in Tab. 8. Our method establishes a new state-of-the-art on all key metrics, achieving the highest average Success Rate (33.12%) and mIoU (65.12%) among all compared methods. Notably, our performance advantage is most pronounced in complex, high-instance-count scenarios. For $L_4$, $L_5$, and $L_6$ levels, our method significantly outperforms all baselines with a Success Rate of **28.12%**, **23.12%**, and **24.38%**, respectively. This demonstrates the robust scalability of our hierarchical architecture to maintain both layout and identity control in intricate scenes. While some competitors show slightly higher scores on individual metrics

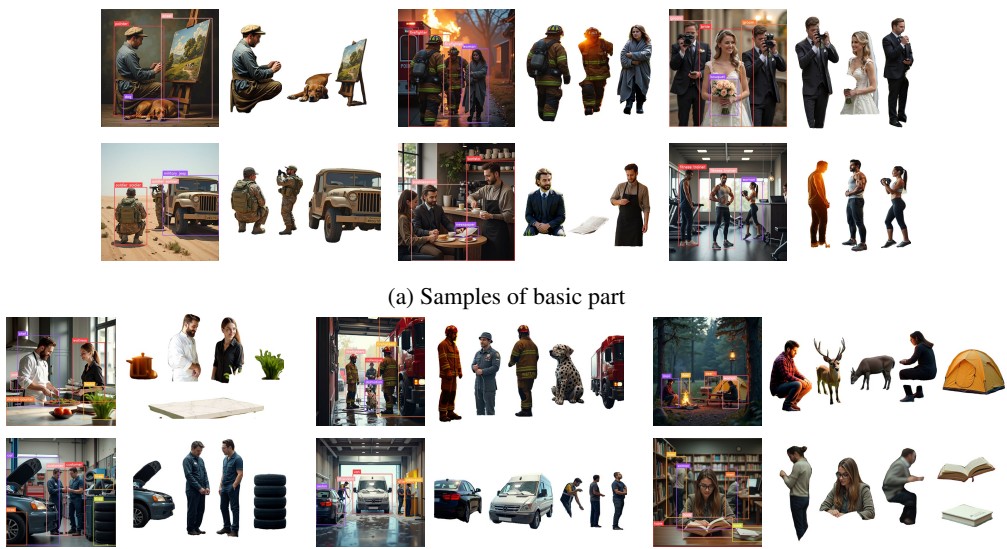

(a) Samples of basic part

(b) Samples of complex part

Figure 13: **Samples of the basic and complex part of IMIG-100K.**

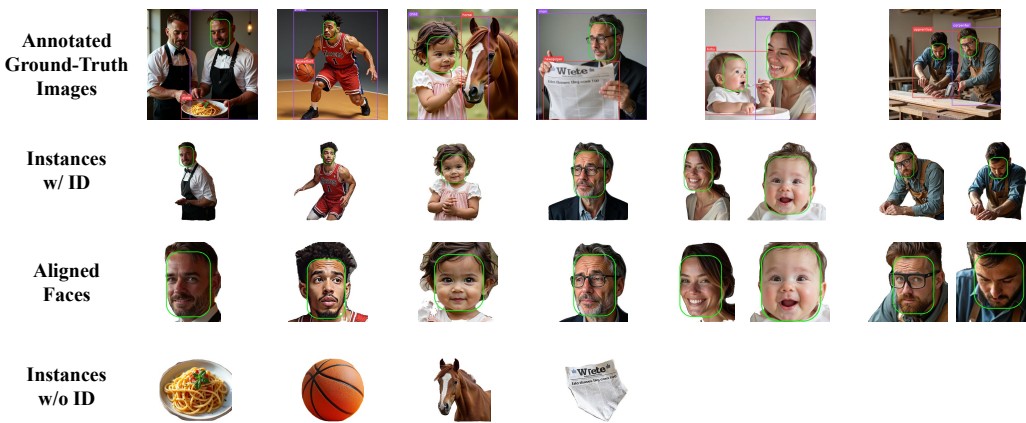

Figure 14: **Samples of the flexible composite part of IMIG-100K.** The faces are annotated in green round bounding boxes.

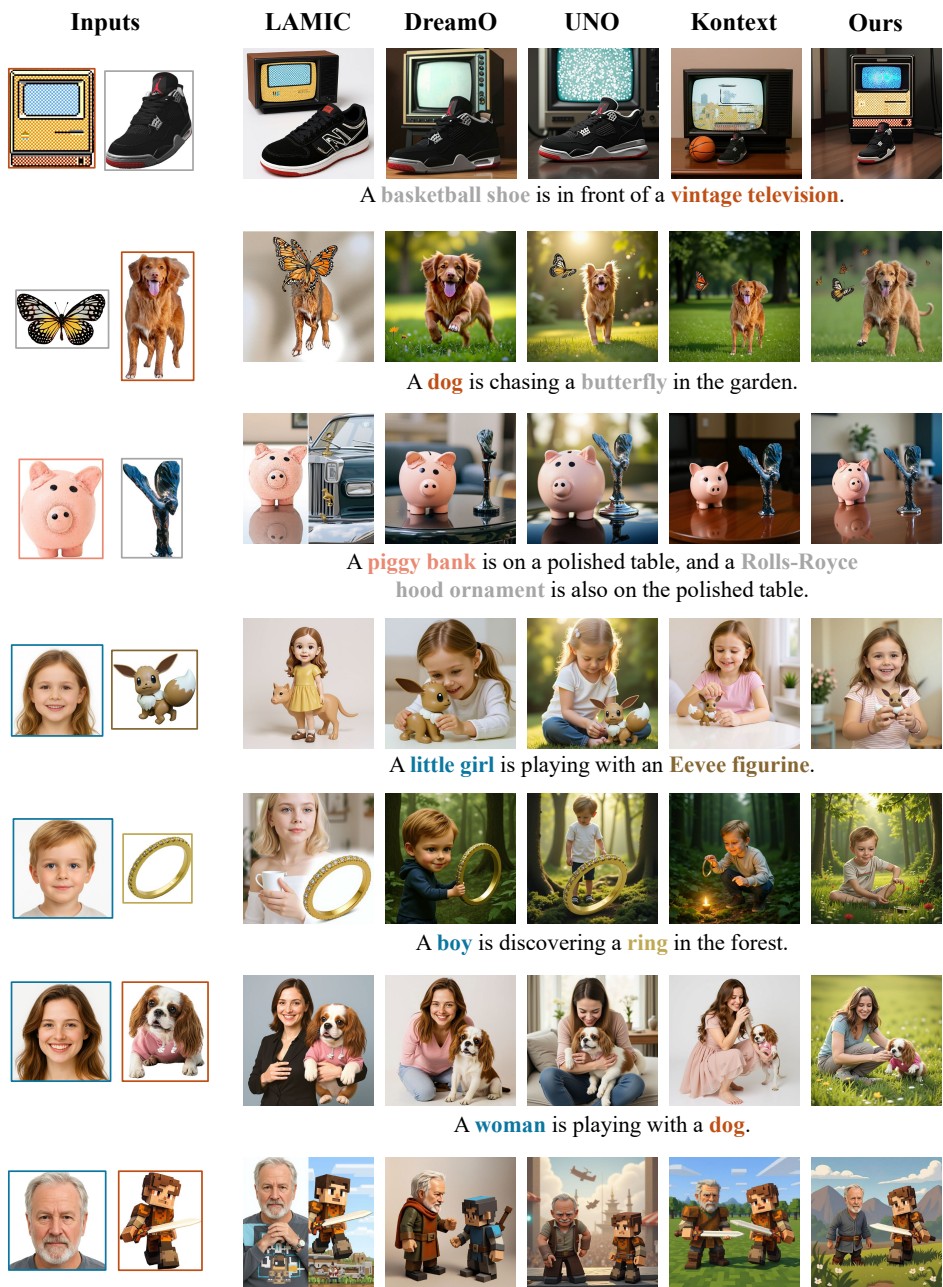

Figure 15: **Qualitative results across FLUX-based methods on LAMICBench++.**

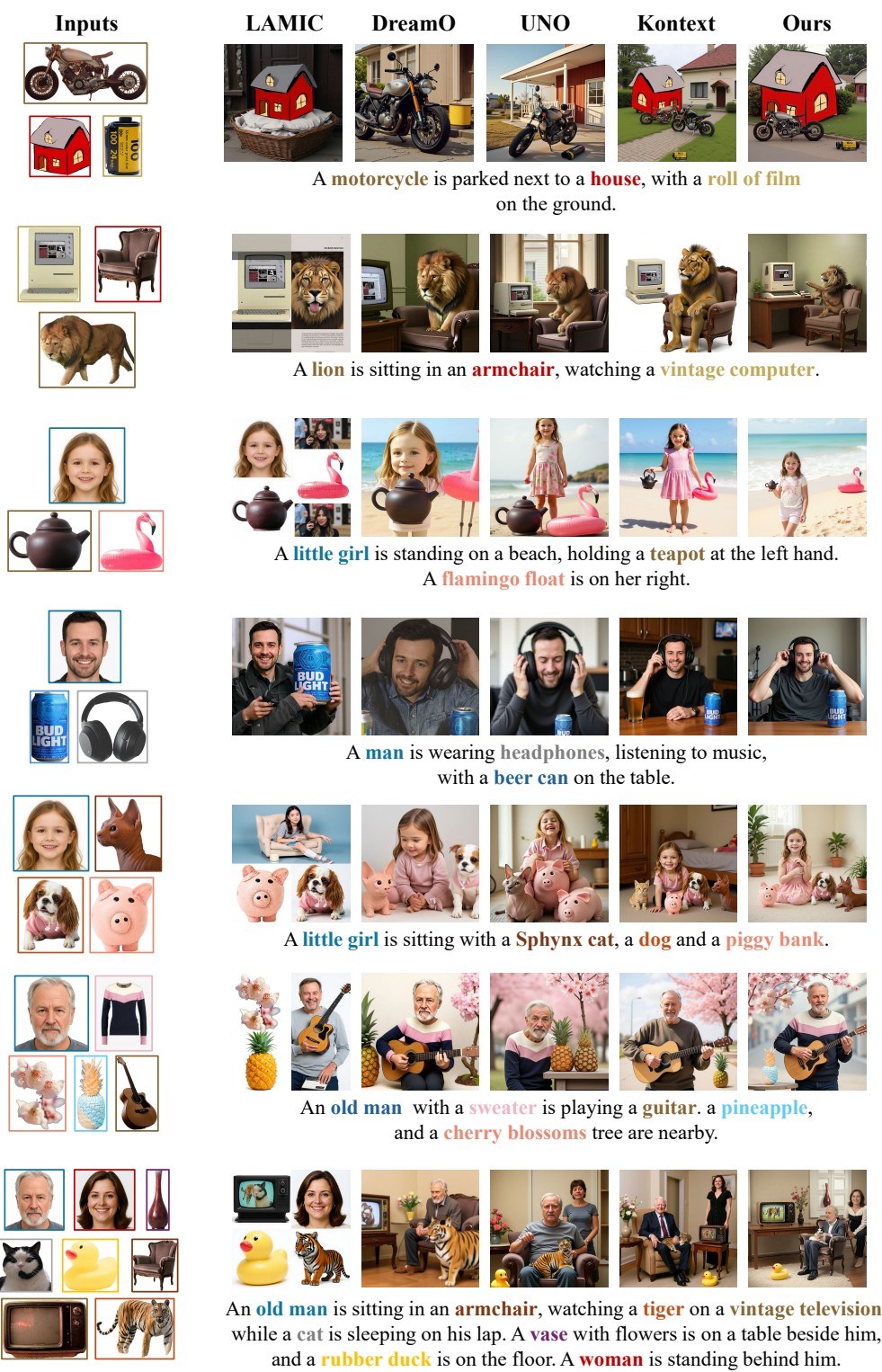

Figure 16: **Qualitative results across FLUX-based methods on LAMICBench++.**

Table 8: **Full quantitative result on COCO-MIG.** According to the count of generated instances, COCO-MIG is divided into five levels: $L_2$, $L_3$, $L_4$, $L_5$, and $L_6$. $L_i$ means that the count of instances needed to generate in the image is $i$.

| Method | Global Clip ↑ | Local Clip ↑ | Success Rate(%) ↑ | | | | | |
|---|---|---|---|---|---|---|---|---|
| | | | Avg | $L_2$ | $L_3$ | $L_4$ | $L_5$ | $L_6$ |
| LAMIC* | 21.82 | 18.71 | 1.25 | 6.25 | 0.00 | 0.00 | 0.00 | 0.00 |
| GLIGEN | 25.21 | 20.90 | 4.25 | 16.88 | 4.38 | 0.00 | 0.00 | 0.00 |
| MS-Diffusion* | 25.50 | 20.77 | 4.50 | 13.75 | 5.62 | 2.50 | 0.62 | 0.00 |
| CreatiLayout | **26.22** | 20.70 | 19.12 | 46.25 | 30.63 | 11.88 | 4.38 | 2.50 |
| 3DIS | 23.72 | 20.40 | 18.88 | 51.88 | 19.38 | 10.62 | 4.38 | 8.12 |
| InstanceDiffusion | 25.77 | **21.91** | 23.00 | 52.50 | 24.38 | 16.88 | 10.62 | 10.62 |
| EliGen | 24.92 | 20.58 | 26.00 | 50.00 | **39.38** | 22.50 | 10.00 | 8.12 |
| MIGC | 26.21 | 21.47 | 27.75 | **53.75** | 34.38 | 21.88 | 11.25 | 17.50 |
| Ours* | 25.86 | 21.87 | **33.12** | 52.50 | 37.50 | **28.12** | **23.12** | **24.38** |

| Method | Instance Success Rate(%) ↑ | | | | | | mIoU ↑ | | | | | |
|---|---|---|---|---|---|---|---|---|---|---|---|---|
| | Avg | $L_2$ | $L_3$ | $L_4$ | $L_5$ | $L_6$ | Avg | $L_2$ | $L_3$ | $L_4$ | $L_5$ | $L_6$ |
| LAMIC* | 13.56 | 28.12 | 19.17 | 13.75 | 9.00 | 9.58 | 21.17 | 31.67 | 25.79 | 20.68 | 18.08 | 18.25 |
| GLIGEN | 29.56 | 41.88 | 31.67 | 27.19 | 27.38 | 27.81 | 27.44 | 37.35 | 29.17 | 25.31 | 26.42 | 25.56 |
| MS-Diffusion* | 28.22 | 37.81 | 33.12 | 28.12 | 25.75 | 24.69 | 34.69 | 41.15 | 36.38 | 34.57 | 32.36 | 33.70 |
| CreatiLayout | 54.69 | 67.19 | 63.33 | 56.09 | 50.25 | 48.96 | 48.96 | 56.32 | 55.38 | 49.42 | 46.22 | 45.28 |
| 3DIS | 55.44 | 71.56 | 61.88 | 55.47 | 48.25 | 52.81 | 49.35 | 61.29 | 53.80 | 49.88 | 44.27 | 47.01 |
| InstanceDiffusion | 60.28 | 71.25 | 61.67 | 59.38 | 57.00 | 59.27 | 54.79 | 65.76 | 57.21 | 53.33 | 51.43 | 53.72 |
| EliGen | 64.12 | 69.69 | **72.50** | 66.56 | 61.62 | 58.54 | 59.23 | 64.61 | 66.10 | 61.59 | 56.74 | 54.50 |
| MIGC | 66.44 | **74.06** | 67.29 | 67.03 | 63.25 | 65.73 | 56.96 | 63.84 | 57.60 | 56.95 | 54.01 | 56.82 |
| Ours* | **69.72** | 70.94 | 69.58 | **72.19** | **68.38** | **68.85** | **65.12** | **66.20** | **66.19** | **66.84** | **63.78** | **64.19** |

like Global Clip (CreatiLayout (Wu et al., 2025c)) or Local Clip (InstanceDiffusion (Wang et al., 2024)), our approach achieves a superior overall balance. The consistently high mIoU scores across all complexity levels and our leading Instance Success Rate on the most challenging cases further validate our model's ability to master both precise instance placement and high-fidelity attribute preservation.

More qualitative results on COCO-MIG are shown in Fig. 17. Our method demonstrates a clear qualitative advantage over existing models on the COCO-MIG benchmark. In the first example, other methods fail to correctly generate the blue vase, with issues ranging from incorrect position to instance merging. Our model, in contrast, precisely renders the vase as intended. Similarly, for the "green potted plant", our approach correctly applies the "green" attribute to the pot, whereas competitors fail to do so. This highlights our superior ability to handle holistic subject identity. Furthermore, while some baselines correctly generate the requested objects in the third and fourth examples, their outputs often lack aesthetic harmony and visual coherence. Our method consistently produces images that are not only accurate but also visually pleasing and well-composed. These results underscore two key advantages of our framework: (1) Compared to other image-guided methods like MS-Diffusion (Wang et al., 2025), our approach offers significant superiority in layout control (2) Our dedicated identity preservation mechanism provides more robust and reliable subject fidelity than the attribute-based control of text-guided methods, particularly in intricate, multi-instance scenes.

## D.3 More Qualitative Results on LayoutSAM-Eval

Additional qualitative results are presented in Fig. 18. Evidently, our method exhibits superior overall visual quality and realism compared to all existing approaches. While other methods (especially those reliant on text-guided layout-to-image generation) struggle with preserving fine-grained attributes—such as the specific text on the building in the first example and the exact color of the man's shorts in the second—our approach faithfully preserves the user's intended details to the greatest extent, excelling in both layout control and attribute binding.

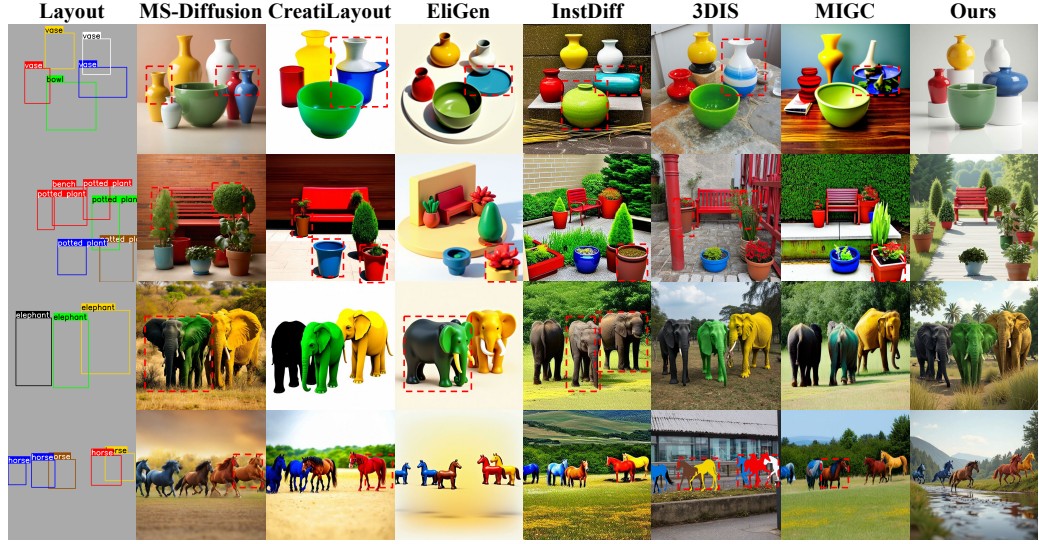

Figure 17: **More qualitative results on COCO-MIG.**

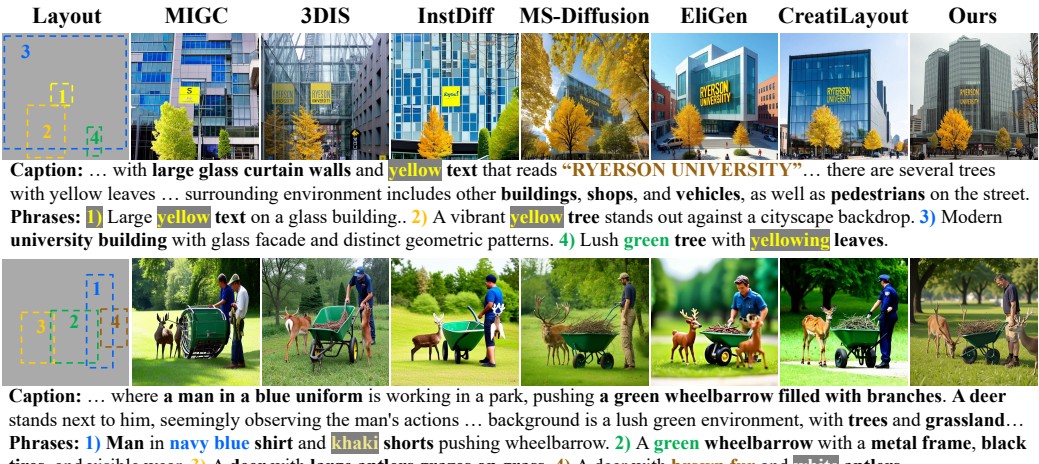

**Caption:** … with **large glass curtain walls** and yellow **text** that reads **"RYERSON UNIVERSITY"**… there are several trees with yellow leaves … surrounding environment includes other **buildings**, **shops**, and **vehicles**, as well as **pedestrians** on the street. **Phrases: 1)** Large yellow **text** on a glass building.. **2)** A vibrant yellow **tree** stands out against a cityscape backdrop. **3)** Modern **university building** with glass facade and distinct geometric patterns. **4)** Lush **green tree** with yellowing **leaves**.

**Caption:** … where **a man in a blue uniform** is working in a park, pushing **a green wheelbarrow filled with branches**. **A deer** stands next to him, seemingly observing the man's actions … background is a lush green environment, with **trees** and **grassland**… **Phrases: 1)** **Man** in **navy blue** **shirt** and khaki **shorts** pushing wheelbarrow. **2)** A **green** **wheelbarrow** with a **metal frame**, **black tires**, and visible wear. **3)** A **deer** with **large antlers grazes on grass**. **4)** A deer with **brown fur** and white **antlers**.

Figure 18: **More qualitative results on LayoutSAM-Eval.**

## E    LIMITATIONS AND FUTURE WORK

While our framework demonstrates state-of-the-art performance in multi-instance generation, it is not without limitations. A primary challenge stems from our model's strong emphasis on identity preservation. When inconsistencies exist between the provided reference images or between the images and the text prompt, our model tends to prioritize maintaining the identities of the reference subjects. This can sometimes lead to a lack of flexibility in adjusting attributes such as lighting, color, or pose, which may compromise the overall visual harmony and text-image consistency of the final output. This trade-off between identity fidelity and contextual flexibility represents an important area for future research. In the future, we plan to explore more dynamic attention mechanisms that can better balance these competing demands, allowing for more flexible style and attribute transfer. Drawing on zero-shot consistency priors (Zhang et al., 2025b), we also aim to incorporate broader generative knowledge to improve the overall harmony of multi-instance scenes. Furthermore, while ContextGen focuses on 2D image synthesis, extending identity-consistent control to 3D assets remains a promising direction (Xu et al., 2024; Shen et al., 2024).

## F    THE USE OF LARGE LANGUAGE MODELS

During the preparation of this manuscript, Large Language Models were used as a general-purpose writing assistant tool. Specifically, LLMs were employed to polish the language and refine the clarity of the text. The authors take full responsibility for the content of the paper.

