# OpenReview forum: "ContextGen: Contextual Layout Anchoring for Identity-Consistent Multi-Instance Generation"
_ICLR.cc/2026/Conference — ICLR 2026 Poster_

### Official Review · Reviewer_LWAe · 2025-10-15

**Soundness:** 1
**Presentation:** 3
**Contribution:** 1
**Rating:** 2
**Confidence:** 4

**Summary:**

This paper introduces ContextGen, a novel Diffusion Transformer-based framework designed for multi-instance image generation, a task that requires precise control over the layout and identity of multiple subjects. The authors identify key challenges in existing methods, namely inadequate position control, weak identity preservation, and a lack of high-quality training data. To address these, they propose a model guided by both layout and reference images, featuring two main technical contributions: Contextual Layout Anchoring (CLA) for spatial control and Identity Consistency Attention (ICA) for identity preservation. Furthermore, they introduce a new large-scale dataset, IMIG-100K, with detailed layout and identity annotations. The experimental results show that ContextGen achieves state-of-the-art performance, outperforming existing open-source and proprietary models on several benchmarks. The strength of the paper lies in the successful integration of these components to produce a highly effective system. However, the work is weakened by concerns about the novelty of its individual methodological components and the conclusiveness of the ablation studies used to validate them.

**Strengths:**

The primary strength of this work is its empirical success. The authors have successfully engineered a complete system that achieves a new state-of-the-art in the complex task of layout-controlled, multi-instance image generation. The combination of a large, well-curated dataset (IMIG-100K), a strong foundation model (FLUX.1-Kontext), a tailored architecture (CLA and ICA), and a robust fine-tuning strategy (including DPO) has proven to be highly effective. The quantitative results across multiple benchmarks, particularly the comprehensive comparison in Table 1, demonstrate a clear advantage over a wide range of baseline methods in overall performance. This indicates that the holistic approach and the specific combination of techniques are well-suited for the problem at hand.

**Weaknesses:**

Despite the strong empirical results, the paper has several weaknesses concerning its methodological contributions and the clarity of their validation.

1.  **Limited Novelty in Core Mechanisms:** Beyond the ICA mechanism, the core methodological contributions appear to be adaptations of existing techniques. Contextual Layout Anchoring (CLA) seems to be a general strategy for multi-reference generation where each reference's attention is constrained, a concept that is not entirely new. The idea of using a composite layout image as an explicit input is interesting, but the paper lacks a specific ablation study to demonstrate its effectiveness compared to a baseline without it. Similarly, the use of Direct Preference Optimization (DPO) is an application of an existing fine-tuning technique. While Table 4 shows it is effective for this task, it does not represent a novel methodological contribution from this work.

2.  **Unconvincing Ablation Study for ICA:** The novelty of the Identity Consistency Attention (ICA) mechanism is the paper's most significant claim, but its validation in Table 3 is not fully convincing. While the final proposed model (applying ICA to the middle blocks) achieves the highest average score and IDS, the performance differences among the last four configurations are marginal and likely within the noise margin. The configuration that completely omits ICA (row beginning with "ITC = 90.26") performs almost as well, which raises questions about whether ICA provides a truly significant or necessary improvement. Furthermore, there is a large, unexplained gap in the IDS metric between the first three configurations and the last four. The paper does not offer an explanation for this pattern, making it difficult for the reader to understand the specific properties or impact of the ICA mechanism. Overall, the ablation study does not provide compelling evidence for the effectiveness of ICA.

3.  **Overstated Contribution of the Dataset:** While the effort in creating the IMIG-100K dataset is substantial, the claim that it is the "first large-scale, hierarchically-structured image-guided multi-instance generation dataset" is questionable. The pipeline described—using generative models to create images and segmentation models to extract references—has become a common practice in the field, with similar efforts seen in works like OmniGen2, Qwen-Image, and MS-Diffusion. As such, while the dataset is a valuable asset, its creation does not constitute a significant research innovation.

4.  **Confounded Baseline Comparisons:** The main results in Table 1 are difficult to interpret due to a lack of controlled variables. The proposed method benefits from a stronger base model (FLUX), a larger and task-specific training dataset (IMIG-100K), and an additional DPO tuning stage, whereas many baselines (e.g., MS-Diffusion) use older, less capable base models and different data. This makes it impossible to attribute the performance gains specifically to the novel CLA and ICA mechanisms. While it is understood that re-implementing all baselines under identical conditions is costly, this confounding of variables limits the scientific insight that can be drawn from the results. The paper demonstrates that a better system can be built with better components, but it fails to provide clear evidence for the isolated impact of its core architectural innovations.

**Questions:**

1.  Could you provide an ablation study that demonstrates the specific benefit of using the composite layout image as an additional input? For instance, a comparison with a model variant that receives only the noise image and the individual reference images, without the pre-composited layout image. This would help clarify the contribution of this specific design choice.

2.  Could you provide a more detailed analysis of the ICA ablation results in Table 3? Specifically, please elaborate on why the performance variations are so small among the last four configurations and justify the final choice. More importantly, please explain the large and systematic performance gap (especially in the IDS metric) between the first three configurations and the last four. What does this reveal about the properties and effectiveness of the ICA mechanism?

---

> ### Author Response · Authors · 2025-11-25
>
> We sincerely appreciate the reviewer for the careful reading of our work and welcome the critical perspective. We provide our responses below.
>
> > **Q1:** Ablation on Composite Layout Image Input:
>
> **A1**: We thank the reviewer for his observation. However, we believe this ablation study is **already addressed** in **the original submission (Sec 4.4, Tab. 3)**, where we've clearly specified around **Line 419 at the caption of the table** that the **Gray Line** denotes the method without CLA (i.e., removing the composite layout image input). The results indicate a significant performance drop **across all key metrics** when the composite layout image is removed, which **validates the effectiveness of CLA** and **the importance of the composite layout image** as an additional input.
>
> > ***A Note on Clarifying CLA and ICA Distinction***
> >
> > Before addressing the subsequent comments, we would like to respectfully clarify the **distinction** between **CLA** and **ICA**, as there appears to be **a divergence in interpreting the functions of these two mechanisms** in the current feedback.
> >
> > Upon reviewing the comments in **Weakness 1 by the reviewer**, we noticed that:
> >
> > 1. The function described in the first part–*"**Contextual Layout Anchoring (CLA)** seems to to be a general strategy ... where each **reference's attention is constrained**..."*– is more accurately describing the mechanism of **ICA**.
> > 2. The next part–*"The idea of using **a composite layout image** as an explicit input is interesting..."*– exactly identifies the **core innovation** of **CLA**.
> >
> > To clarify their distinct functions in the ContextGen framework:
> >
> > - **Contextual Layout Anchoring (CLA)** is the mechanism that utilizes the **composite layout image** as an additional contextual input for layout guidance.
> > - **Identity Consistency Attention (ICA)** is the mechanism that uses **constrained attention** to enhance identity preservation.
> >
> > For ease of verification, we sincerely encourage the reviewer to check our description in **original submission (Sec. 1, Line 76-81, and the detailed mechanisms in Sec. 3.2)**. We hope to highlight that **a correct understanding** of CLA's and ICA's unique roles is **important for evaluating our contribution and experiment results**.
> >
> > Our following responses to the **remaining comments** (especially **Weakness 1,2** and **Question 2**) proceed based on the **original definitions** of CLA and ICA. We are more than glad to make any further clarifications if needed.

---

> ### Author Response · Authors · 2025-11-25
>
> > **Q2:** Detailed Analysis of ICA Ablation Results:
>
> **A2**: We appreciate the reviewer's request for a deeper analysis of the ICA ablation results presented in Tab. 3. The table's result reveals a **potential functional specialization** within the FLUX-DiT layers, where performance is dependent on the **placement of the attention mechanism**:
>
> - **F-19: Strategy to Minimize Modality Interference:** Prior work **CreatiLayout**[1], which extending **only the Front-19 blocks** (Double Stream Blocks) for layout-to-image generation, has shown a**powerful control in both spatial and attribute**. And as stated this work, **no further modification is made to the other blocks** (Single Stream Blocks, i.e., Mid-19 and Back-19 blocks) in order to **reduce the potential interference among modalities** (i.e., text, layout, and image). It does make sense in considering the Front-19 blocks contains **huge amount of parameters** than the others. Adding ICA to these blocks is fairly enough, while applying ICA to more blocks lead to side effects like identity distortions. That **aligned with our observation** that $F+M+B$ and $F+B$ strategies underperform others.
> - **M-19: Critical Role in Attribute Binding:** Prior work **DreamRenderer**[2] demonstrates through ablation study that Mid-19 blocks play a relatively more critical role in attribute binding. Our ablation study validate the finding, for applying ICA to Mid-19 blocks yields the best performance in our task on benchmark score.
> - **F-19 Protection and Similar Performance:** The four strategies, $F$, $M$, $M+B$, and no ICA, yield **relatively close performance scores** because they all **avoid posing further disruptions** to the Front-19 blocks. And we can derive from this result that these four strategies do not have a very significant difference in overall performance.
> - **B-19: Detrimental Impact on Final Refinement:** Applying ICA **solely to the Back-19 blocks** proves detrimental to performance, which is also aligned with the ablation study in **DreamRenderer**[2]. For the Back-19 blocks serve as **final and sensitive post-processing layers** in each noise prediction step. Introducing attention mask **exclusively at this late stage** without any modifications to the earlier blocks will affect the **refinement process**, leading to problems like artifacts and blurring identity.
>
> We choose to **apply ICA to Mid-19** from the last four strategies with very close score for the following reasons:
>
> - **Indispensability** of ICA: The analysis in **Appendix B5 of the revised version** has revealed the indispensability of ICA.
>
> - **Performance**: Applying ICA to the Mid-19 blocks demonstrates an advantage of **overall benchmark score** over other strategies.
>
> - **Efficiency**: Considering that the computational cost of applying the attention mechanism scales with both the number of blocks and the block capacity (such as F-19 blocks with much denser parameters), we selected the Mid-19 configuration for a **computationally efficient approach** to achieve better performance.
>
> We hope this detailed analysis clarifies the observed performance patterns and the rationale behind our final choice regarding the ICA mechanism.

---

> ### Author Response · Authors · 2025-11-25
>
> **We also want to respectfully respond to the weaknesses raised by the reviewer:**
>
> > **Weakness1:** Limited Novelty in Core Mechanisms:
>
> We welcome reviewer's questions on the novelty of our core mechanisms. Given the **potential for misunderstanding** regarding the **definitions of CLA and ICA**, as addressed in the ***preceding Note*** in **Answer 1**, we respectfully detail the novelty and specific contributions of **each mechanism** as follows:
>
> - **Novelty of CLA:** As clarified in our preceding *Note*, the CLA is not about constrained attention (which belongs to ICA), but the strategic integration of the **Composite Layout Image** as an explicit contextual input, which **the reviewer generously remarked as "interesting"**. Its novelty lies in using **image-level layout information** (the Composite Layout Image) as a contextual input for **layout control** in multi-instance generation, which has **not been explored in prior works**. As stated in our **Answer 1**, the **ablation study** available in the **original submission** has already demonstrated its effectiveness.
> - **Specialization of ICA**: We respectfully agree that the idea of **constrained attention** has been explored in existing backbones like **Stable Diffusion**. However, the innovation of ICA lies in its unique **architectural specialization** and **systemic role**. ICA is specifically designed and optimized for the **FLUX-DiT architecture**, focusing **detail identity preservation**. Crucially, ICA works **in  synergy with** CLA, forming a **layered and unified** solution that effectively handles the multi-instance generation challenges.
> - **Application Novelty of DPO:** We really appreciate the reviewer's recognition of DPO's effectiveness in our task. We would like to respectfully highlight that while DPO is an existing technique, its **application to multi-instance generation** is has not been explored in prior works. The integration of DPO into our work represents a **meaningful advancement** in fine-tuning strategies, as it is specifically tailored to enhance the performance of multi-instance generation models by optimizing the **fidelity-flexibility trade-off**.
>
> Thus, we believe that each of our core mechanisms and their **synergistic integration** in ContextGen, represents a **novel contribution** to the field of multi-instance generation. We sincerely hope this clarification addresses the reviewer's concern regarding the novelty of our work.

---

> ### Author Response · Authors · 2025-11-25
>
> > **Weakness2:** Unconvincing Ablation Study for ICA:
>
> We appreciate the reviewer's thoughtful analysis of ICA's role as evidenced in Tab. 3. We appreciate the opportunity to clarify this aspect of our work:
>
> - **ICA as Co-Essential Contribution:** We wish to clarify that our paper **does not position ICA as the most significant contribution**. As stated in the paper, **both** CLA and ICA are **essential components** of ContextGen, each addressing different aspects of the multi-instance generation challenge.
>
> - **Unique Advantage Beyond Benchmarks:** While the benchmark scores among the top configurations are subtle, we believe that ICA provides **advantages that the metrics may not fully capture**. Our extended analysis in **revised version (Appendix B.5)** specifically elaborates on ICA’s ability to **handle overlapping instances**—as **initially noted** in the **original submission (Sec. 1, Line 73)**. Although ICA's impact on benchmark scores is limited, this functionality—visually verified in **Fig. 11 of the revised version**—leads to **observable gains** in **identity and detail fidelity**. We sincerely refer the reviewer to these additional results for validation.
>
> - **Rationale Detailed in Structural Analysis:** In **response to Question 2** above, we now provide a detailed analysis of the ICA ablation results and the rationale behind the observed performance patterns.
>
> The combination of the **detailed structural analysis** and the **additional qualitative evidence** confirms that ICA is a indispensable component for enhancing identity and detail fidelity. We hope this explanation will address the reviewer's concern.

---

> ### Author Response · Authors · 2025-11-25
>
> > **Weakness3:** Overstated Contribution of the Dataset:
>
> We welcome the reviewer's scrutiny regarding the originality and contribution of our dataset. We would like to utilize this opportunity to clarify the contribution and uniqueness of our dataset IMIG-100K in more detail:
>
> - **Primary Claim - Dataset Uniqueness and Completeness:** Acknowledging the reviewer's perspective, we **respectfully maintain** that our dataset is indeed the first large-scale, hierarchically-structured image-guided multi-instance generation dataset with **layout and identity annotations**, as completely described in **Sec. 1, Line 97 of the original submission**. And no prior work has created a dataset specifically tailored for image-guided multi-instance generation with **all** the essential characteristics including **detailed layout and identity annotations, hierarchical structure, and the scale that IMIG-100K offers**.
> - **Distinct Synthesis Methodology and Data Structure:** Regarding the synthesis pipeline, we **acknowledge the utility** of generative and segmentation models in dataset construction. However, we would like to politely clarify that our approach is **distinct from simple image-to-reference extraction**. As detailed in the **original submission (Sec. 3.3, Line259)**, our methodology employs **two distinct synthetic pipelines**. Crucially, this includes a complex, multi-stage approach where **individual reference are generated first** and then **composed into the target scene** by generative models. This specialized, reverse process significantly enhances the data's subject diversity and transformation capability, underscoring our methodological innovation and data structure specialization.
>
> - **Comparison with Existing Datasets:** The prior works mentioned by the reviewer, while valuable, have **limitations** for **Multi-Instance Generation (MIG)** tasks and have **significant differences** compared to our contribution. To be specific:
>
>   - **OmniGen-Series/X2I-Dataset: Limited Instance Count and Missing Layout.** The subject-driven part in X2I Dataset[3] of OmniGen Series primarily relies on an *MLLM-based cross-verification pipeline* for gathering **real-world pairs**. This **collection methodology** inherently results in data with a **limited number of instances (mostly fewer than 3 subjects)**. Crucially, this dataset also lacks the detailed spatial or layout annotations necessary for robust spatial-aware generation.
>   - **Qwen-Image: General-Purpose Scope.** Though the dataset of Qwen-Image[4] has included synthetic data, it's a more **comprehensive image generation** dataset without any **specific focus or tailoring** for multi-instance generation tasks. Its synthetic data pipeline is primarily designed to **enhance text rendering** and **image editing consistency**, making it general-purpose rather than specialized for robust spatial-aware multi-instance generation.
>   - **MS-Diffusion: Source Limitation and Restricted Variation.** Dataset of MS-Diffusion[5], consists of images collected from **real-word video frames** via a *cross-verification* pipeline, suffers from an inherent limitation: its low identity matching rate across frames forces the **direct use of ground-truth parts as references** (as stated in its paper), which restricts meaningful variation and transformations between reference and target subjects.
>
>   We summarize the key distinctions in the following table:
>
>   | Dataset | Data Source | Construction Method | Primary Focus | Limitation for MIG | Layout/Identity Annotations |
>   | - | - | - | - | - | - |
>   | X2I(OmniGen Series) | Mostly Real-world (Web Images) | MLLM Cross-verification | Subject-Driven Customization | Limited instance count | No/Missing |
>   | Qwen-Image | Mixed | Comprehensive Data Pipeline | General T2I | Not specialized for MIG | No/Missing |
>   | MS-Diffusion | Real-world (Video Frames) | Cross-verification | Image-Guided MIG | Restricted subject variation | Yes (Layout), Limited Identity |
>   | **Ours** | **Synthetic** | **Dedicated Dual Pipeline** | Image-Guided MIG | None (Specialized for this task) | **Yes (Essential, Detailed)** |
>
>   As mentioned above, **none** of these works utilize a **dataset composition** or **construction pipeline** similar to ours. Our methodology is built on the **specialized manufacturing** and **rigorous post-processing** of synthetic data. The resulting data structure is **uniquely tailored for multi-instance generation** through reinforcing **complex subject compositions** and providing **essential layout and identity annotations**.
>
> Thus, we believe IMIG-100K constitutes a **significant contribution** to the field, providing both a **valuable training resource** and establishing **a scalable methodology for dataset synthesis**. We kindly refer the reviewer to **Appendix C of the revised version**, which contains a brief survey and discussion on existing multi-instance datasets, along with further details of our proposed dataset.

---

> ### Author Response · Authors · 2025-11-25
>
> > **Weakness4:** Confounded Baseline Comparisons:
>
> We appreciate the reviewer's concern regarding the **confounding factors** in our baseline comparisons. However, we believe that the performance improvements demonstrated by ContextGen are **not solely attributable to the capability of FLUX backbone and large-scale dataset**. Instead, they stem from our **proposed architectural innovations**—CLA and ICA—and their effective integration with the FLUX architecture. We provide the following evidence to support this claim:
>
> - **Comparison Across FLUX-Based Methods:** Tab. 1 clearly shows in quantitative metrics that our method outperforms other **FLUX-based** methods, including **vanilla FLUX.1-Kontext**, **LAMIC (based on FLUX.1-Kontext)**, **UNO and DreamO (both based on FLUX.1-Dev)**. The review can check more qualitative comparisons across these FLUX-based methods in **Appendix D.1 (Fig.15-16, for two full pages) of the revised version**.
>
> - **Implementation of the MS-FLUX Competitor:** To better show the isolated impact of our core architectural innovations, we have implemented a competitor **"MS-FLUX"** by applying MS-Diffusion's core mechanism–Grounding Resampler–to FLUX.1-Kontext backbone (*in response to the reviewer's point regarding the older base model of MS-Diffusion*). The **implementation details (including the core code) and benchmark results** are included in **Appendix B.7 of the revised version**. (The benchmark results are also summarized below.)
>
>     | Method       | SR    | I-SR  | mIoU  | IDS   | IPS   |
>     |--------------|-------|-------|-------|-------|-------|
>     | MS-FLUX      | 0.38  | 7.30   | 18.71 | 3.20  | 60.22 |
>     | MS-Diffusion | 4.50  | 28.22 | 34.69 | 9.06  | 69.75 |
>     | Ours         | 33.12 | 69.72 | 65.12 | 32.72 | 76.32 |
>
> - **Validation of Our Architectural Effectiveness:** The quantitative results on benchmarks show that MS-FLUX suffers from **severely poor identity preservation** coupled with **a near-complete degradation of layout control**. This failure case demonstrates that despite FLUX.1-Kontext's strong capability, not all methods **(even those working well on other backbones)** can effectively address the challenges of Multi-Instance Generation on the FLUX architecture. And our proposed components are **particularly effective** in this regard.
>
> Therefore, while recognizing the difficulties in controlling all external variables, our combined evidence–**performance against other FLUX-based methods** and the **comparative results of MS-FLUX**–shows the effectiveness of our core architectural innovations **beyond the backbone and dataset advantages**. We hope this explanation adequately address the reviewer's concern.
>
> References:
>
> [1] Zhang, et al. "CreatiLayout: Siamese Multimodal Diffusion Transformer for Creative Layout-to-Image Generation". ICCV 2025. [https://arxiv.org/pdf/2412.03859](https://arxiv.org/pdf/2412.03859)
>
> [2] Zhou, et al. "DreamRenderer: Taming Multi-Instance Attribute Control in Large-Scale Text-to-Image Models". ICCV 2025. [https://arxiv.org/pdf/2503.12885](https://arxiv.org/pdf/2503.12885)
>
> [3] Xiao, et al. "X2I-subjecr-driven". Hugging Face Datasets. [https://huggingface.co/datasets/yzwang/X2I-subject-driven](https://huggingface.co/datasets/yzwang/X2I-subject-driven)
>
> [4] Wu, et al. "Qwen-Image Technical Report". ArXiv 2025. [https://arxiv.org/pdf/2508.02324](https://arxiv.org/pdf/2508.02324)
>
> [5] Wang, et al. "MS-Diffusion: Multi-subject Zero-shot Image Personalization with Layout Guidance". ICLR 2025. [https://arxiv.org/pdf/2406.07209](https://arxiv.org/pdf/2406.07209)

---

### Official Review · Reviewer_4W69 · 2025-10-29

**Soundness:** 3
**Presentation:** 3
**Contribution:** 3
**Rating:** 6
**Confidence:** 3

**Summary:**

The paper advances multi-instance generation (MIG) with not only precise layout control, but identity preservation. To simultaneously achieve  these two goals, the authors devise CLA and ICA to manipulate attention masks for better identity and location understanding. Moreover, they also propose a large-scale dataset to facilitate future research in this field. Extensive experiments on three large-scale benchmarks demonstrate their method's efficacy.

**Strengths:**

(1) The proposed dataset, IMIG-100K, has a significant meaning to the IMIG field.

(2) The overall training process is clear and could be easily followed.

(3) Comparison with ``Closed-Source Commercial Models'' in Tab.1 strongly demonstrates the efficacy of proposed method.

**Weaknesses:**

Major:
(1) The inference process is not clear on pure "layout-to-image" benchmarks. Based on Fig. 2, the ref images seems to be mandatory for model's inference. However, the included layout-to-image benchmark (COCO-MIG and LayoutSAM-eval) only provide layout and instance captions. How to perform inference on these benchmarks with yoour model? Is it possible to infer your model without any reference images? Please clarify this.

(2) Instance-Wise Position Indexing is one of the contributions for this work, but I cannot see any ablation study about this.

(3) In Fig.7, the generated image exhibits noticeable artifacts (crab legs on the bag) with DPO. Therefore, it is necessary to report metrics about image quality (FID or IS). The authors can compute FID and report the result on LayoutSAM-eval as CreatiLayout has also reported FID on this benchmark.


Minor:
(1) Typo: Tab.1 #L338, "FLUX" is mis-spelled.

Ref.
[a] CreatiLayout: Siamese Multimodal Diffusion Transformer for Creative Layout-to-Image Generation. In ICCV'25.

**Questions:**

Please refer to "weaknesses"

---

> ### Author Response · Authors · 2025-11-25
>
> We sincerely appreciate the reviewer for the careful reading of our work and insightful comments. We provide our responses below.
>
> > **Q1:** Inference process on pure Layout-to-Image benchmarks:
>
> **A1**: We really appreciate the reviewer's careful examination of our evaluation setup on the pure Layout-to-Image benchmarks (COCO-MIG and LayoutSAM-Eval). As explicitly stated in the **original submission (Tab. 2, Line 380)**, we used **pre-generated images by FLUX.1-Dev** as the reference images for our model on these benchmarks. This allows our Image-Guided model to participate in the evaluation.
>
> The reviewer may raise **a valid concern** that this approach might **unfairly inflate the scores** for attribute control metrics. However, our results demonstrate the superiority of ContextGen even when controlling for this potential factor:
>
> - **Superior Layout Control Under Adverse Conditions**: Even if we disregard attribute-related metrics, our method still achieves leading performance on **layout related metrics** (e.g., SR and mIoU). On the other hand, these two benchmarks enforce a square aspect ratio by **stretching the layout bounding boxes**, resulting in many geometrically challenging shapes. This is actually **unfavorable** for Image-Guided methods, as the aspect ratio of the pre-generated reference images are very likely to **clash with these distorted bounding boxes**. Despite this, our Contextual Layout Anchoring (CLA) mechanism proves highly effective.
> - **Outperforming Image-Guided Methods**: Leaving aside the text-guided methods, our ContextGen **surpasses existing Image-Guided methods** (LAMIC and MS-Diffusion) across all evaluated metrics. This validates that our architectural innovations provide substantial and necessary advancements over prior image-guided SOTA, independent of the reference image acquisition methodology.
>
> Therefore, our approach demonstrates a clear advantage in **layout control** and outperforms prior **Image-Guided SOTA** within this comparison setting. We hope this explanation adequately addresses the reviewer's concern regarding our evaluation setup on pure Layout-to-Image benchmarks.

---

> ### Author Response · Authors · 2025-11-25
>
> > **Q2:** Absence of ablation on position indexing:
>
> **A2**: We appreciate and fully agree with the reviewer's suggestion of including an ablation study on the position indexing mechanism. We have now conducted the ablation study on LAMICBench++ and included the detailed results in **Appendix B.2 (Tab. 5) in revised version**. The **core results** are summarized as follows, demonstrating the necessity of **both $x$ and $y$ offsets**, which are essential for enabling the non-overlapping indexing mechanism. Removing **any offset component** resulted in a significant degradation in identity fidelity.
>
> | Position Indexing | IDS | IPS | AVG |
> |-------------------|-----|-----|-----|
> | w/o $x$ and $y$ offsets | 21.54  | 74.57  | 61.14 |
> | w/o $y$ offset     | 21.01  | 74.02  | 60.71 |
> | w/o $x$ offset     | 26.71  | 74.59  | 62.67 |
> | w/ both offsets    | **32.72**  | **76.32**  | **64.66** |
>
> We hope this additional ablation study can adequately address the reviewer's insightful suggestion.

---

> ### Author Response · Authors · 2025-11-25
>
> > **Q3:** Report on FID metric:
>
> **A3**: We really appreciate the reviewer for this careful observation and insightful comments regarding the potential artifacts introduced by DPO fine-tuning.
>
> - **Quantitative Quality Reporting:** We now report the FID scores on LayoutSAM-Eval along with the Aesthetic scores (AES) on LAMICBench++ under different DPO $\beta$ settings in **Appendix B.3.2 (Tab. 6) of the revised version** (also summarized below).
> - **Limitations of FID on LayoutSAM-eval:** We would like to point out that the FID on LayoutSAM-eval **may not fully reflect the image quality**, not only because FID is less sensitive to **fine-grained image details**, but also because the calculation process necessitates **severe distortion** by resizing the structurally diverse ground-truth images of this benchmark. We add **qualitative results** in **Fig. 8 of the revised version** to better illustrate that the best FID score (DPO $\beta=500$) **does not necessarily** correspond to the best visual quality.
> - **User Study Validation:** To address the limition of FID score and better evaluate the image quality, we conducted a **user study** using samples in LayoutSAM-eval and LAMICBench++. The user study settings and detailed results are also included in **Appendix B.3.2 of the revised version**. The key finding is **aligned with our previous observation** that DPO fine-tuning will enhance the image quality, and a smaller $\beta$ (a more aggressive DPO) generally leads to better quality.
>
>     | DPO $\beta$ | AES               | FID               | User Preference (by user study)  |
>     |-------------|-------------------|-------------------|------------------|
>     | 100         | **57.97**         | 55.97             | **0.54**         |
>     | 250         | 57.58             | 55.60             | 0.37             |
>     | 500         | 57.57             | **55.32**         | 0.16             |
>     | 750         | 57.22             | 55.53             | 0.03             |
>     | 1000        | 57.10             | 55.65             | -0.11            |
>     | 1500        | 56.69             | 55.70             | -0.43            |
>     | w/o DPO     | 54.19             | 55.93             | -0.56            |
>
> We hope these additional results and analyses adequately address the reviewer's concerns regarding image quality and the effects of DPO fine-tuning.

---

### Official Review · Reviewer_CnRS · 2025-10-30

**Soundness:** 3
**Presentation:** 3
**Contribution:** 3
**Rating:** 6
**Confidence:** 4

**Summary:**

The authors introduce ContextGen, a novel Diffusion Transformer (DiT) framework designed to tackle the significant challenges of multi-instance image generation (MIG). The method aims to solve two key problems: poor layout control and the failure to preserve the identities of multiple distinct subjects. The framework integrates two primary technical innovations: 1) a Contextual Layout Anchoring (CLA) mechanism to precisely position objects, and 2) an Identity Consistency Attention (ICA) mechanism that uses reference images to maintain the identity of each instance. Recognizing a lack of training data, the authors also created IMIG-100K, a new large-scale dataset for this task. The paper reports state-of-the-art performance in control precision, identity fidelity, and image quality.

**Strengths:**

- The work's primary strength is its direct and simultaneous attack on the two most critical failure points in multi-instance generation: layout control and identity preservation.
- The paper proposes two clear, novel technical components (CLA and ICA) integrated into a modern DiT framework. The Identity Consistency Attention, in particular, appears to be a sophisticated mechanism tailored specifically for the multi-subject challenge.
- The introduction of the IMIG-100K dataset is a valuable contribution in its own right. A large-scale, hierarchically-structured dataset with detailed layout and identity annotations addresses a major resource gap and will be beneficial for the wider community.

**Weaknesses:**

- Fidelity vs. Generative Diversity Trade-off: The model appears to struggle with the balance between preserving identity and generating novel depictions. As suggested by the results (e.g., Figure 4, rows 1 and 3), the generated instances often exhibit a "copy-paste" artifact, adhering so rigidly to the reference images that they lack meaningful diversity. This suggests a potential overfitting to the reference context and limits the model's true generative flexibility.
- Sub-optimal Performance on Key Metrics: Despite the SOTA claims, the method does not achieve top performance across all evaluated metrics. This performance gap may be linked to the newly introduced IMIG-100K dataset. By relying heavily on synthetic data, the model may not learn the full complexity and nuances of real-world image distributions, which could be limiting its generalization and robustness. Incorporating more authentic, real-world images into the training curriculum might be necessary to close this gap.

**Questions:**

See Weaknesses.

---

> ### Author Response · Authors · 2025-11-25
>
> We really appreciate the reviewer for the positive feedback and insightful comments. We provide our responses below.
>
> > **Q1:** Fidelity vs. Generative Diversity Trade-off:
>
> **A1**: We agree that our model—owing to its strong anchoring and attention mechanism—will **prioritize maintaining identity**, which may lead to a "copy-and-paste" appearance, especially when it is provided with a **minimal or simple text prompt**. We would like to make clarifications as follows:
>
> - **Flexibility Conditioned by Prompt Richness:** When the prompt is enriched with complex descriptions of **inter-subject interactions** or **dynamic scenes**, the model demonstrates **significant adaptability** by **modifying postures and attributes** of the subjects to comply with the textual context. This proves that the model does not rigidly "transfer" the references but is capable of high generative flexibility.
>
> - **Qualitative Demonstration of Adaptability:** We have included **new qualitative analysis and results** in the **Appendix B.4 (Fig. 10) of the revised version** that specifically showcase the model's adaptive generation capabilities under varied and complex prompts. We encourage the reviewer to examine this appendix for a detailed assessment of the model's flexibility.
>
> We hope these clarifications and additional results can address the reviewer's concerns regarding the trade-off between identity fidelity and generative diversity.

---

> ### Author Response · Authors · 2025-11-25
>
> > **Q2:** Sub-Optimal Performance on Key Metrics linking to Dataset:
>
> **A2**: We sincerely appreciate the reviewer's careful examination of our quantitative results. While readily acknowledging that the synthetic dataset IMIG-100K **may not contain the full complexity and nuances** of real-world image distributions, we would like to offer the following clarifications regarding performance and its link to our dataset:
>
> - **Overall Performance Prioritization**: We acknowledge that ContextGen **has not achieved** the absolute best score across all generalized metrics. But our SOTA claim is based on the **superior performance in the core metrics** defining the Multi-Instance Generation (MIG) task: **Identity Consistency** (e.g. IDS/IPS in LAMICBench++) and **Layout Adherence** (e.g. SR/mIoU in COCO-MIG). And we are convinced that there's a **fundamental trade-off** between layout/identity fidelity and generative diversity/quality (*just as the reviewer has pointed out in Q1*). By **focusing on these core metrics** and also **strike a balance with other generalized metrics**, we believe ContextGen effectively addresses the primary challenges of MIG while still delivering competitive overall performance.
>
> - **IMIG-100K Generalization Capability**: We **respectfully disagree with** the suggestion that the IMIG-100K dataset **is responsible for** performance gaps. Our synthetic dataset was created with a strict, multi-stage filtering process focused on image **quality and diversity**. We encourage the reviewer to check more details and sample images of our dataset in **Appendix C.2 and Fig.12-14 of the revised version**. More importantly, ContextGen's robust performance on established public benchmarks with **exclusively real-world data**, demonstrates the **strong generalization and robustness** of the model trained on our data. The performance on these external benchmarks confirms that the model has learned the **necessary complexity** for real-world applications.
>
> - **Synthetic Data as a Valid Methodology**: Existing methods like **UNO**[1] and **XVerse**[2] have already evolved synthetic data for training subject-driven generation models. The practice in these prior works have demonstrated that synthetic datasets, when carefully constructed and filtered, can **effectively train models** that generalize well to **real-world scenarios**. We encourage the reviewer to check our brief survey and discussion on existing multi-instance datasets in **Appendix C.1 of the revised version**.
>
> We believe that our achievement of SOTA in the specialized metrics relevant to ContextGen's **core objective**, combined with **strong generalization evidence**, validates both the model's design and the utility of the IMIG-100K dataset.
>
> References:
>
> [1] Wu, et al. "Less-to-More Generalization: Unlocking More Controllability by In-Context Generation". ICCV 2025. [https://arxiv.org/pdf/2504.02160](https://arxiv.org/pdf/2504.02160)
>
> [2] Chen, et al. "XVerse: Consistent Multi-Subject Control of Identity and Semantic Attributes via DiT Modulation". NeurIPS 2025. [https://arxiv.org/pdf/2506.21416](https://arxiv.org/pdf/2506.21416)

---

### Official Review · Reviewer_pU4T · 2025-11-03

**Soundness:** 3
**Presentation:** 3
**Contribution:** 3
**Rating:** 6
**Confidence:** 4

**Summary:**

This paper addresses the challenging task of multi-instance image generation (MIG), which requires simultaneous control over the layout of multiple objects and the preservation of their distinct identities. The authors propose ContextGen, a novel Diffusion Transformer (DiT) based framework. The framework features two core innovations: 1) Contextual Layout Anchoring (CLA), which uses a "composite layout image" as context to precisely anchor object positions; and 2) Identity Consistency Attention (ICA), a novel attention mechanism that leverages reference images to maintain high-fidelity identity for multiple instances. Furthermore, to address the lack of high-quality training data in this domain, the authors have constructed and introduced a large-scale, hierarchically-structured dataset, IMIG-100K. The experimental evaluation is extensive. On three different benchmarks (LAMICBench++, COCO-MIG, and LayoutSam-Eval), ContextGen achieves state-of-the-art performance, outperforming not only all open-source models but also top-tier proprietary models (like GPT-4o and Nano Banana) on key identity preservation metrics.

**Strengths:**

1. Empirical Results: As mentioned, the paper's experimental results are its greatest strength. It comprehensively outperforms all competitors (both open-source and proprietary) on three distinct and challenging benchmarks. The ability to handle multiple (>3) subjects, shown in Tables 1 and 4, is a capability that other models clearly lack.
2. High-Quality Dataset Contribution: The release of IMIG-100K  is a major contribution. The authors not only created a dataset but thoughtfully designed it in three subsets (Basic, Complex, Flexible) to support the model's progressive training curriculum .
3. Architectural Design: The hierarchical attention design is excellent. Rather than blindly applying attention masks, the authors discovered through experiments (Table 3) that different layers of the DiT indeed have functional specializations—CLA is applied to the first/last blocks for structure, while ICA is applied to the middle blocks for identity details.

**Weaknesses:**

1. Confusing DPO Ablation: This is the only confusing part of the paper. Table 3  shows the best (non-DPO) model (F+M+B) achieving an AVG score of 64.66. However, in Table 4, the "w/o DPO" baseline has an AVG score of only 62.55. Both scores are presumably from LAMICBench++, but they do not match. The authors claim DPO provides a benefit (62.67 vs 62.55) , but this seems to be a degradation compared to the 64.66 score in Table 3. The authors need to clarify this discrepancy.
2. Reliance on Base Model: This work is a LoRA fine-tune on a very strong (and until recently, non-public) base model, FLUX.1-Kontext. This is not a flaw, but rather an efficient research methodology. However, it does mean the success is built, in part, on the inherent in-context capabilities of the Kontext model. The authors are transparent about this in Appendix A.1 , which is good.
3. Dataset Limitations: The IMIG-100K dataset is synthetic, meaning it may not fully capture the entire complexity and long-tail distribution of real-world imagery.

**Questions:**

Please referring Weakness.

---

> ### Author Response · Authors · 2025-11-25
>
> We are grateful to the reviewer for the constructive and encouraging feedback. Our detailed responses are provided below.
>
> > **Q1:** Confusing DPO Ablations:
>
> **A1**：We really appreciate the reviewer's careful reading of our quantitative results. We clarify the confusion regarding the DPO ablation results in Tab. 4 compared to other tables (Tab. 1 - 3) as follows:
>
> - **Different LoRA Rank:** The DPO ablation in Tab. 4 was conducted on **LoRA Rank 256** (refer to **Line 454 in the original submission**). The results in other tables (Tab. 1 - 3) were conducted on **LoRA Rank 512** (refer to **Line 283 in the original submission**) with DPO $\beta = 1000$.
> - **Resource Constraint Strategy:** The difference of LoRA Rank metioned above is due to the **limited computational resources** during experiments. DPO fine-tuning with LoRA Rank 512 requires **significantly larger GPU memory usage** (**$\geq$ 80GB**), which exceeds our internal available resources (NVIDIA A100 80G). Therefore, we first conducted **the DPO ablation on LoRA Rank 256** to identify the optimal $\beta$ value, and then applied this setting to the **final model with LoRA Rank 512** on **borrowed external resources** (NVIDIA PRO6000 96G).
> - **Qualitative Validation on Final Model**: We have validated the effectiveness of DPO fine-tuning on the **final LoRA Rank 512 model**. The additional qualitative comparisons are included in **Appendix B.3.3 (Fig. 9) of the revised version**. And the results validate that DPO fine-tuning consistently improves the image quality of the final model as well.
>
> We hope this explanation clarifies the confusion.

---

> ### Author Response · Authors · 2025-11-25
>
> > **Q2**: Reliance on Base Model:
>
> **A2**: We sincerely appreciate the reviewer's acknowledgement that utilizing FLUX.1-Kontext is an effective research methodology. And we are grateful for the reviewer's positive remark regarding our transparency in Appendix A.1. We would like to clarify several points regarding **our reliance on the FLUX backbone**.
>
> - **Backbone Selection:** we wish to clarify that the selection of the FLUX variant mentioned in Appendix A.1 is **not predicated on maximizing base-model capacity**. Rather, our intention was to identify the FLUX variant **best suited for** integrating **our specific contextual attention mechanisms**.
>
> - **FLUX Family's Inherent Limitations:** we agree that FLUX backbone provides a strong foundation. However, despite FLUX family's strength, their inherent architecture is **insufficient** for **achieving highly-customized, identity-consistent** Multi-Instance Generation. Specifically:
>
>   - The FLUX series lacks dedicated mechanisms for **explicit and robust layout control**.
>
>   - Its identity preservation capability is **not robust enough** to maintain high fidelity when simultaneously generating multiple distinct instances.
>
> - **ContextGen's Superiority within FLUX Family:** Our components—Contextual Layout Anchoring (CLA) and Identity Consistency Attention (ICA)—systematically resolve the backbone's inherent limitations. As demonstrated by our quantitative results in Tab. 1, the full ContextGen framework achieves a highly significant performance gain over other **FLUX-based methods**, including **vanilla FLUX.1-Kontext, LAMIC (based on FLUX.1-Kontext), UNO and DreamO (both based on FLUX.1-Dev)**, particularly in the multi-instance IDS and IPS metrics. This proves that our methods successfully transform the general-purpose FLUX architecture **into a specialized, SOTA solution** for identity-consistent Multi-Instance Generation task. In order to further illustrate our generation performance compared to other FLUX-based methods, we have added **extensive qualitative comparisons** in **Appendix D.1 (Fig.15-16, for two full pages) of the revised version**. We encourage the reviewer to check these comparisons to better appreciate the effectiveness of our proposed components.
>
> Therefore, we believe that the performance improvements of ContextGen are **not merely due to the choice of FLUX backbone**, but rather stem from our **architectural contributions** that effectively address the unique challenges of the Multi-Instance Generation task.

---

> ### Author Response · Authors · 2025-11-25
>
> > **Q3**: Dataset Limitations:
>
> **A3**: We fully agree with the reviewer that synthetic datasets, including our IMIG-100K, **may not perfectly** capture the full complexity and long-tail distribution of real-world data. We are committed to exploring real-world applications and generalization capabilities in future work.
>
> However, we would like to clarify that, owing to critical limitations of existing real-world data for training MIG models, the creation of IMIG-100K is a necessary strategic choice for this work. We explain our rationale as follows:
>
> - **Scale and Diversity Limitations of Real-World Datasets:** Obtaining the necessary scale of data with explicit multi-subject identity references **(especially the ground-truth images and reference images are both real-world data)** is prohibitively hard and complex. Existing real-world datasets lack the diversity and volume to support effective training on our task. We encourage the reviewer to refer to our brief survey and discussion on **existing multi-instance datasets** in **Appendix C.1 of the revised version**.
>
> - **High Quality of Our Dataset:** Our dataset has undergone **strict filtering**, including checks based on **human preference scoring** and **quality assessment models**. This ensures the quality of the synthetic data, making it suitable for training high-performance models. The reviewer can check **more sample images of our dataset** in **Appendix C.2 and Fig.12-14 of the revised version**.
>
> - **Established Precedent of Synthetic Data:** The use of high-quality synthetic data is an established and effective methodology for training subject-driven generation models, as demonstrated by **prior works** such as **UNO**[1] and **XVerse**[2].
>
> - **Generalization Capability of ContextGen:** The strong performance of ContextGen on **real-world benchmarks** (all the three benchmarks exclusively use real-world images) demonstrates the model's good generalization capability, despite being trained on a synthetic dataset.
>
> Therefore, we believe that our proposed dataset is a **justified choice** for this work and this task, effectively addressing the limitations on **data scale, diversity and quality** which are critical for training high-performance MIG models.
>
> References:
>
> [1] Wu, et al. "Less-to-More Generalization: Unlocking More Controllability by In-Context Generation". ICCV 2025. [https://arxiv.org/pdf/2504.02160](https://arxiv.org/pdf/2504.02160)
>
> [2] Chen, et al. "XVerse: Consistent Multi-Subject Control of Identity and Semantic Attributes via DiT Modulation". NeurIPS 2025. [https://arxiv.org/abs/2506.21416](https://arxiv.org/abs/2506.21416)

---

### Author Response · Authors · 2025-12-02
**Rebuttal Summary for Submission 8501**

### Dear Area Chairs,

We sincerely appreciate your consideration of our submission, and we are grateful to all the reviewers for their constructive feedback. Their comments have been invaluable in helping us refine and improve our work. Below we summarize the contributions and response to the reviewers' concerns.

### **Summary of Contributions**

1. **Novel Framework.** We propose **ContextGen**, a novel DiT-based architecture tailored for multi-instance image generation tasks, powered by two core mechanisms: **Contextual Layout Anchoring (CLA)** for efficient layout control, and **Identity Consistency Attention (ICA)** for detail identity preservation.

2. **SOTA Performance.** ContextGen achieves state-of-the-art performance over existing methods **(both open-source and proprietary)** for image generation challenges of layout control and identity preservation (praised by **All the Reviewers**).

3. **Large-Scale Dedicated Dataset.** We introduce **IMIG-100K**, the **first large-scale synthetic dataset** specifically designed for multi-instance image generation tasks with **detailed layout and identity annotations**, along with its full construction pipeline (commended by **Reviewer pU4T, CnRS, and 4W69**).

4. **Extensive Ablations and Performance Enhancements.** We carefully design the architecture and ablate the effectiveness of our attention mechanism on different DiT layers (highlighted by **Reviewer pU4T**), and we strategically employ DPO fine-tuning to improve the equilibrium between flexibility and fidelity, demonstrating its effectiveness for this task (acknowledged by **Reviewer LWAe**).

### **Summary of Revisions**

We have carefully addressed all concerns raised by the reviewers:

**Reviewer pU4T:**

- **DPO Ablation Settings**: Confirmed the different LoRA settings used  and validated DPO's effectiveness on the final model with additional results **(Appendix B.3.3, Fig. 9)**.
- **Backbone Reliance**: Added extensive analyses and qualitative comparisons within FLUX-family methods  **(Appendix D.1, Fig. 15-16)**, demonstrating the superiority of our mechanism beyond the backbone.
- **Dataset Limitations**: Conducted a brief survey of existing datasets and provided a discussion to clarify the necessity of our synthetic dataset **(Appendix C.1)**. Added more details and samples to illustrate the dataset's construction and quality **(Appendix C.2, Fig. 12-14)**.

**Reviewer CnRS:**

- **Generative Diversity Concerns**: Analyzed and added more qualitative results to demonstrate the generative diversity of our model **(Appendix B.4, Fig. 10)**.
- **Performance & Dataset Justification**: Clarified the SOTA criteria based on core MIG metrics and justified the generalization of our synthetic dataset by its quality **(Appendix C.2)**.

**Reviewer 4W69:**

- **Benchmark Settings**: Reiterated the experimental settings for pure layout-to-image benchmarks, which were already stated in the **original submission (Tab. 2, Line 380)**.
- **Ablation on Position Indexing**: Added an ablation study to validate the necessity of our non-overlapping position indexing strategy **(Appendix B.2, Tab. 5)**.
- **Quality Assessment of DPO**: Provided a more detailed assessment, including user study results, for the image quality improvement brought by DPO fine-tuning **(Appendix B.3.2, Tab. 6)**.

**Reviewer LWAe:**

- **Ablation Study of CLA**: Re-emphasized the ablation results of CLA, which were already presented in the **original submission (Tab. 3, Line 419)**.
- **Analysis of ICA Ablation**: Provided a detailed structural analysis **(Appendix B.6)** and qualitative results **(Appendix B.5, Fig.11)** of the ICA module to reveal its working mechanism on FLUX-DiT layers and demonstrate its effectiveness in preserving detail identity.
- **Novelty of Mechanism**: Clarified the distinct novelty of each core mechanism (CLA, ICA, and DPO fine-tuning) within ContextGen.
- **Contribution of Dataset**: Clarified the critical differences between IMIG-100K and existing datasets mentioned by the reviewer, justifying its novelty and necessity for the MIG task **(Appendix C)**.
- **Fair Baseline Comparison**: Implemented and benchmarked a new competitor, MS-FLUX, by applying the mechanism of MS-Diffusion to FLUX-Kontext backbone **(Appendix B.7, Tab. 7)**, thus isolating and explaining the effectiveness of our proposed mechanism.

All changes are **highlighted in blue** in the revised paper. We believe the substantial revisions have thoroughly addressed all reviewer concerns, and we are confident the revised manuscript constitutes a significantly improved contribution. We hope the Area Chair finds our responses satisfactory.

Sincerely,

The Authors of Submission 8501

---

### Meta-Review · Area_Chair_bUXs · 2025-12-17

**Summary:**

In general, the paper’s strong results are widely acknowledged, but the decision is pulled toward concerns about:
1. Missing/insufficient ablations for claimed components:
The ablation study is not comprehensive making it hard to know: Since the proposed method combines several powerful factors at once—a very strong FLUX backbone, a large task-specific dataset (IMIG-100K), and extra fine-tuning (incl. DPO)—while many baselines use weaker backbones and different data/training configs

2. Incomplete reporting around DPO and image-quality metrics (e.g., artifacts, missing/limited quality measures).

**Reviewer Concerns:**

In general, reviewers are concerned about the ablation study of this paper is not comprehensive and in addition most of the novelties are from other papers and authors just borrow the idea.

**Reviewer Scores:**

three with 6
One with 2

---

### Decision · Program_Chairs · 2026-01-26

Accept (Poster)